# Precision Targeting in Metastatic Prostate Cancer: Molecular Insights to Therapeutic Frontiers

**DOI:** 10.3390/biom15050625

**Published:** 2025-04-27

**Authors:** Whi-An Kwon, Jae Young Joung

**Affiliations:** 1Department of Urology, Hanyang University College of Medicine, Myongji Hospital, Goyang 10475, Republic of Korea; 2Department of Urology, Urological Cancer Center, National Cancer Center, Goyang 10408, Republic of Korea

**Keywords:** metastatic prostate cancer, precision medicine, PARP inhibitors, PSMA-targeted therapy, tumor microenvironment

## Abstract

Metastatic prostate cancer (mPCa) remains a significant cause of cancer-related mortality in men. Advances in molecular profiling have demonstrated that the androgen receptor (AR) axis, DNA damage repair pathways, and the *PI3K/AKT/mTOR* pathway are critical drivers of disease progression and therapeutic resistance. Despite the established benefits of hormone therapy, chemotherapy, and bone-targeting agents, mPCa commonly becomes treatment-resistant. Recent breakthroughs have highlighted the importance of identifying actionable genetic alterations, such as *BRCA2* or *ATM* defects, that render tumors sensitive to poly-ADP ribose polymerase (PARP) inhibitors. Parallel efforts have refined imaging—particularly prostate-specific membrane antigen (PSMA) positron emission tomography-computed tomography—to detect and localize metastatic lesions with high sensitivity, thereby guiding patient selection for PSMA-targeted radioligand therapies. Multi-omics innovations, including liquid biopsy technologies, enable the real-time tracking of emergent AR splice variants or reversion mutations, supporting adaptive therapy paradigms. Nonetheless, the complexity of mPCa necessitates combination strategies, such as pairing AR inhibition with PI3K/AKT blockade or PARP inhibitors, to inhibit tumor plasticity. Immuno-oncological approaches remain challenging for unselected patients; however, subsets with mismatch repair deficiency or neuroendocrine phenotypes may benefit from immune checkpoint blockade or targeted epigenetic interventions. We present these pivotal advances, and discuss how biomarker-guided integrative treatments can improve mPCa management.

## 1. Introduction

Prostate cancer (PCa) is one of the most commonly diagnosed malignancies in men globally, with more than 1.4 million new cases and more than 375,000 deaths annually [1]. North America, Europe, and Australia report particularly high incidence rates, reflecting the effects of genetic factors, lifestyle, and screening practices [2]. While localized or locally advanced disease can often be managed with surgery or radiation therapy, a notable percentage of patients present with metastatic disease or progress to metastasis, notably to the bones, lymph nodes, lungs, or liver [1,2]. Once PCa metastasizes, the 5-year survival rate remains poor, especially for metastatic castration-resistant PCa (mCRPC) [3].

Over the past two decades, next-generation androgen receptor (AR) pathway inhibitors, taxane-based chemotherapy, bone-targeted therapies, and prostate-specific membrane antigen (PSMA)-directed radioligand therapies have resulted in increased survival rates. Nevertheless, treatment resistance frequently arises, driven by AR splice variants, lineage plasticity, and the upregulation of survival pathways [4]. The emergence of precision oncology has been driven by high-throughput DNA/RNA sequencing, single-cell omics, and other advanced technologies [5]. Landmark projects (TCGA, SU2C/PCF Dream Teams) have identified alterations in DNA damage repair (DDR) genes (*BRCA2*, *ATM*, and *CHEK2*), in the *PI3K/AKT/mTOR* pathway (often via the loss of *PTEN*), and in AR amplifications/mutations [6]. The clinical relevance of these alterations is evident in the success of poly-ADP ribose polymerase (PARP) inhibitors (olaparib, rucaparib) in the treatment of patients with mCRPC harboring DDR mutations [7,8,9].

AR splice variants, particularly AR-*V7*, have emerged as biomarkers of resistance to AR-targeted therapies [10]. PSMA positron emission tomography-computed tomography (PET-CT) enables the earlier detection of micrometastatic disease and guides PSMA-targeted radioligand therapy [11].

However, metastatic PCa remains highly heterogeneous [12,13]. Clonal evolution and an immunosuppressive tumor microenvironment (TME) impede durable responses, emphasizing the need for TME reprogramming [14,15,16,17,18,19].

Multidisciplinary strategies integrating genomic data, advanced imaging, adaptive trials, and real-world evidence aim to refine patient stratification, identify actionable targets, and optimize therapy sequences [20,21,22,23,24].

As PCa management becomes increasingly complex, because of the rapid expansion of precision oncology tools, novel therapeutic agents, and dynamic insights into the TME, an up-to-date synthesis of these advancements is vital. A timely review will provide clinicians and researchers with the necessary context to integrate emerging data into evidence-based practice, will help overcome resistance mechanisms, and will realize the promise of truly personalized, potentially transformative care.

This review discusses the molecular underpinnings of disease progression, key diagnostic and stratification methods (e.g., next-generation sequencing [NGS], liquid biopsies), current and emerging therapeutics (AR inhibitors, PARP inhibitors, immunotherapies, radioligand therapies), and future directions for TME modulation and artificial intelligence (AI)-driven approaches. Ultimately, this comprehensive synthesis aims to improve outcomes and explore the potential for long-term disease control—even cure—in carefully defined molecular subsets.

## 2. Molecular Pathophysiology of Metastatic Prostate Cancer

Several key molecular alterations drive the progression and therapeutic resistance of metastatic PCa (mPCa), including AR amplification/splice variants, *PTEN* loss, and DDR defects. These alterations are summarized in Table 1 along with their clinical implications and potential therapeutic avenues.

### 2.1. Centrality of the AR Axis

#### 2.1.1. Historical Underpinnings and Core AR Functions

In the 1940s, Charles Huggins provided seminal evidence that prostate tumors regress under surgical castration or estrogen therapy, thus establishing androgen dependence as a fundamental driver of PCa progression [25]. This discovery remains the cornerstone of therapy for advanced or high-risk PCa, for which androgen-deprivation therapy (ADT) continues to be the mainstay. Although early studies could not fully harness modern molecular techniques, they laid the groundwork for understanding the pivotal “testosterone–dihydrotestosterone (DHT)–AR” signaling axis in both normal and malignant prostate cells.

AR is a ligand-dependent transcription factor comprising an N-terminal transactivation domain (NTD), a DNA-binding domain (DBD), a hinge region, and a ligand-binding domain (LBD). In normal prostate cells, testosterone is converted to DHT, which binds the AR to the LBD, triggers a conformational change, and facilitates nuclear translocation. Once in the nucleus, AR binds to the androgen response elements and recruits cofactors (e.g., p300/CBP and SRC-1) to regulate genes that promote proliferation, survival, and secretory functions.

Early work with first-generation anti-androgen drugs (e.g., flutamide) showed temporary growth control, but soon revealed a rapid onset of resistance driven by compensatory alterations within the AR signaling pathway [26,27]. These findings underscore the remarkable adaptability of the AR pathway, and highlight the clinical challenges associated with CRPC. While confirming that the central role of AR is sufficient for clinical effect, advanced molecular and genomic technologies now enable deeper insights into how AR signaling becomes dysregulated or adapts to therapeutic pressure.

Early in vivo tumor regrowth assays have demonstrated the importance of AR signaling; however, they lack the robust molecular stratification required to capture heterogeneity among different patient subgroups [28]. Consequently, historical clinical trials have provided only partial insights into the full spectrum of AR mutations, splice variants, and crosstalk with growth factor pathways. Over time, the integration of genomic, transcriptomic, and proteomic techniques has yielded more nuanced patient-specific treatment approaches, continually raising the pivotal clinical and scientific question of how best to inhibit AR-related tumor progression [29].

#### 2.1.2. Routes to AR Reactivation Under Therapeutic Pressure

During prolonged ADT for PCa, strong selective pressure within a low-androgen environment favors the survival of tumor cell clones that maintain AR activity, ultimately leading to CRPC. In this setting, AR gene amplification and overexpression enable tumors to respond to even minimal levels of androgens or weak agonists, while mutations in the receptor’s LBD, such as T877A or F876L, can convert certain anti-androgen factors into partial agonists that promote tumor growth [30]. Additionally, growth factor pathways, such as the *IGF-1R* and *EGFR* pathways, can activate AR in a ligand-independent manner, and enzymes such as *CYP17A1* or *AKR1C3* facilitate intratumoral androgen synthesis, providing an alternative route for tumor cells to circumvent low-androgen conditions. The effectiveness of abiraterone stems in part from its ability to inhibit *CYP17A1* and block endogenous androgen production. Furthermore, AR splice variants (e.g., AR-*V7*) remain constitutively active, even without ligand binding, rendering them resistant to therapies targeting the LBD, such as enzalutamide [31]. The fact that second-generation AR-directed drugs (e.g., enzalutamide, abiraterone) confer survival benefits to patients with CRPC underscores the persistent dependence of most PCa on the AR axis, even after ADT failure [30]. Multiple mechanisms contribute to AR reactivation in CRPC, including ligand-binding domain mutations, AR amplification, and the emergence of constitutively active splice variants. Figure 1 illustrates these pathways, highlighting how each can lead to resistance against second-generation AR-targeted therapies.

Research is now extending toward novel AR inhibition approaches, including NTD inhibitors [32] and proteolysis-targeting chimeras (PROTACs) that degrade both full-length AR and splice variants [33]. Combination regimens that merge AR-targeted therapies with PI3K/AKT inhibitors, epigenetic modulators, or immunotherapeutic agents are being explored to address the multifaceted resistance mechanisms of CRPC [34,35]. While metastatic tumor samples and cell line models have been critical to understanding how AR reactivation occurs, they often lack the ability to fully replicate the complexity and heterogeneity of the clinical TME, which includes immune and stromal factors [36]. This limitation highlights the need for refined multi-omic profiling and more sophisticated ex vivo and organoid models that can more accurately reflect individual patient contexts [36,37].

Ultimately, AR signaling remains a major determinant of long-term PCa control, and unraveling patient-specific AR mutations and TME interactions may reveal predictive markers for therapeutic resistance and guide personalized combination regimens. Epigenetic modifications, such as methylation or acetylation, may further regulate AR splice variant expression, offering fresh targets and diagnostic biomarkers for treatment-resistant disease [29]. A deeper understanding of AR reactivation via multiple pathways is essential for developing layered inhibition strategies and precision approaches, both of which are crucial for overcoming treatment resistance and improving patient outcomes [35].

### 2.2. PARP Inhibitors and the Synthetic Lethality Paradigm 

#### 2.2.1. Historical Context and Foundational Insights

PARP enzymes are central to the repair of single-stranded DNA breaks. The concept of synthetic lethality, initially demonstrated in *BRCA1/2*-mutant breast and ovarian cancers, provides a new perspective on how defective homologous recombination (HR) renders tumor cells exceptionally vulnerable to *PARP* disruption [38]. Early evidence of this vulnerability, which showed a pronounced reliance on *PARP* when HR pathways were compromised, shifted the focus of researchers in the broader DDR field to investigate whether similar liabilities exist in other malignancies, including for PCa [39].

The demonstration of synthetic lethality has moved beyond the classical one-gene/one-enzyme paradigm, showing that blocking a compensatory repair route, such as PARP, can selectively kill HR-deficient cells. These foundational insights have spurred extensive research into DDR pathways, revealing that *BRCA1/2* mutations are only one facet of a larger network of potential repair defects [40].

#### 2.2.2. Core Molecular Mechanisms in Prostate Cancer

Prostate tumors may harbor a spectrum of DDR alterations beyond those in *BRCA1*/*2*, including those in *ATM* and other HR-associated genes [41]. Under normal conditions, *PARP* mediates the repair of single-strand breaks. However, when *PARP* activity is inhibited, unrepaired single-strand breaks collapse into double-strand breaks (DSBs). Cells equipped with robust HR machinery typically resolve DSBs. By contrast, HR-deficient cells fail to complete repair, leading to catastrophic DNA damage, and ultimately, cell death [42]. *BRCA2*- or *ATM*-deficient prostate tumors exhibit heightened sensitivity to PARP inhibitors via synthetic lethality. Figure 2 summarizes the core mechanism by which *PARP* blockade leads to irreparable DSBs in DDR-deficient cells, ultimately causing cell death.

While *BRCA2*- and *ATM*-mutated prostate tumors often exhibit more pronounced susceptibility to *PARP* blockade, some cases without *BRCA* mutations also display partial responses [43]. The reasons for this are not completely understood, suggesting that the DDR landscape in PCa is more intricate than initially assumed [40]. Additional genetic or epigenetic defects, potentially in Fanconi anemia genes, other HR components, or even in replication stress-related pathways, may confer comparable vulnerabilities. Ongoing research highlights the complex network of repair proteins that modulates the response of tumors to DNA damage. Variations in the TME, mutational load, and epigenetic regulation may further shape DDR competence, explaining partial or atypical responses [41].

#### 2.2.3. Adaptive and Resistance Mechanisms

Despite the clear link between HR defects and increased *PARP* sensitivity, prostate tumors can acquire resistance over time. One well-characterized route involves reversion mutations, in which previously inactivated HR-associated genes (e.g., *BRCA2*) regain partial or full function. In other cases, alternative repair pathways, such as non-homologous end joining or translation DNA synthesis, may be upregulated, compensating for the blocked *PARP* axis and enabling tumor cells to survive. The restoration of HR capacity undercuts the premise of synthetic lethality and complicates long-term control [44]. Additionally, the inherent redundancy of the DDR network underscores the need for a comprehensive molecular profile for each tumor, rather than relying on a single mutational marker [45].

#### 2.2.4. Research Methodologies and Knowledge Gaps

Large-scale genomic profiling, including whole-exome and whole-genome sequencing, along with multi-omics approaches that integrate transcriptomics, proteomics, and epigenomics, have significantly advanced our understanding of DDR perturbations in PCa [6]. Nonetheless, important gaps remain in the literature [43]. Biomarker panels tend to center on *BRCA1/2* and a limited set of DDR-related genes, potentially overlooking less common or functionally redundant pathways [45]. In addition, prostate tumors often exhibit marked heterogeneity among patients, and across different metastatic sites within a single patient, complicating the interpretation of DDR defects [6]. A further challenge arises when tumors show DDR-related abnormalities primarily at the functional level—such as replication stress—rather than through clear genomic alterations, such that they may evade detection by standard gene panels.

A critical appraisal of existing evidence demonstrates that large-scale, multi-institutional genomic studies have consistently linked *BRCA2* and *ATM* mutations with DDR dysfunction, yet data on noncanonical DDR alterations frequently come from small cohorts or exploratory analyses, limiting their broader applicability [45]. While some researchers have emphasized *BRCA* gene alterations as the primary drivers of PARP inhibitor sensitivity, others have advocated a more comprehensive perspective that accounts for epigenetic factors and secondary DNA repair pathways [46]. This debate underscores the inherent complexity of tumor biology, and highlights the pressing need for holistic approaches that integrate genomics, epigenomics, and functional assays to better characterize and target DDR dysregulation in PCa.

#### 2.2.5. Forward-Looking Perspectives in Molecular Pathophysiology

Next-generation technologies are transforming our understanding of DDR and synthetic lethality in prostate cancer by moving beyond traditional gene mutation models. These approaches collectively enrich the classification of prostate tumors, revealing vulnerabilities that expand the paradigm of synthetic lethality. In parallel, the historical success of *PARP* inhibition in *BRCA*-mutant breast and ovarian cancers underscores how targeting single-strand break repair can be lethal for cells in which HR is already compromised, a principle now applied to PCa, with a growing list of DDR defects [47]. The discovery of reversion mutations and alternative repair routes highlights the complexity of the DDR landscape, driving efforts to refine molecular classifications and rational therapeutic strategies [48]. As advanced genomic techniques, functional assays, and computational modeling continue to evolve, they promise to illuminate new mechanisms of DNA repair vulnerability in PCa, ultimately enabling more precise interventions and improved clinical outcomes [47].

### 2.3. TMPRSS2–ERG Fusions and Oncogenic Transcription Factors (Molecular Pathophysiology Focus)

#### 2.3.1. Historical to Current Understanding

The discovery of *TMPRSS2*–*ERG* fusions marked a milestone in deciphering how androgen-responsive promoters drive aberrant transcriptional programs in PCa. Early studies established that *ERG*, which belongs to the *ETS* transcription factor family, is overexpressed when placed under the regulatory control of the AR-responsive *TMPRSS2* promoter [49]. This rearrangement was initially considered a distinct molecular subtype of PCa.

Over the past decade, numerous studies have examined how *TMPRSS2*–*ERG*-positive tumors intersect with other molecular subtypes; however, the precise prognostic significance of these fusions remains debatable. Historically, the discovery of these fusions has provided a novel example of how AR signaling can be co-opted, revealing the broader principles of oncogenic transcription in PCa. Although *ERG* fusions are frequently detected, their direct role in driving tumor aggressiveness remains controversial, with conflicting reports on correlations with clinical outcomes [49].

#### 2.3.2. Mechanistic Insights

*ERG* modulates numerous gene expression pathways implicated in epithelial–mesenchymal transition (EMT) and genomic instability [50]. Under the control of *TMPRSS2*, *ERG* is regulated by androgens, reinforcing a feed-forward loop in which AR signaling indirectly augments oncogenic transcription [51]. Notably, co-alterations, most prominently the loss of *PTEN*, can synergize with *ERG* overexpression, potentiating pathways that promote proliferation, invasion, and possibly metastasis [50]. *TMPRSS2*–*ERG* fusions result in androgen-responsive overexpression of *ERG*, promoting invasive and pro-oncogenic gene programs. Figure 3 schematically depicts how the *TMPRSS2* promoter drives *ERG* transcription, thereby linking AR signaling to enhanced tumor invasiveness.

A critical challenge lies in the absence of well-defined *ERG* domains that are amenable to direct pharmacological inhibition. This molecular architecture has complicated efforts to suppress the oncogenic activity of *ERG*. Consequently, *ERG* co-regulators, such as BRD4 and DNA-PK, have garnered attention as alternative therapeutic interventions, although these directions remain under investigation at the mechanistic level, rather than as established clinical strategies. The ability of *ERG* to modulate genes associated with invasion, EMT, and DNA repair demonstrates its integration with other oncogenic signals in a context-dependent manner [51]. Furthermore, *PTEN* deficiency often co-occurs with *ERG* overexpression, accentuating its pro-tumorigenic effects [50].

#### 2.3.3. Methodological Limitations

In many early cohort studies, patients were not uniformly profiled for additional molecular events beyond *TMPRSS2*–*ERG* fusion. This lack of comprehensive molecular annotation has confounded efforts to parse the specific contribution of *ERG* to tumor aggressiveness, especially in the presence of the loss of *PTEN* or other concurrent mutations [52].

*TMPRSS2*–*ERG*-positive cell lines and xenograft models often fail to capture the TME and intrapatient heterogeneity of a disease. Furthermore, the reliance on overexpression systems can exaggerate or misrepresent physiological *ERG* levels, complicating efforts to draw definitive conclusions [53]. Although correlative data suggest an association between *ERG* fusion and certain tumor phenotypes, the strengths of these associations vary widely. Differing assay sensitivities and patient selection criteria, such as Gleason score distributions and prior therapies, further contribute to the contradictory findings regarding prognosis [54].

#### 2.3.4. Clinical or Scientific Significance

From a molecular pathophysiological standpoint, *TMPRSS2*–*ERG* fusions illustrate how a normal physiological pathway, AR signaling, can be subverted to drive oncogenic transcription. Notably, *ERG* overexpression upregulates invasion- and EMT-related genes, and potentially modulates *PI3K*/AKT signaling, DDR networks, and epigenetic regulators. This interplay underscores the complexity of PCa biology, as single genetic events, such as *TMPRSS2*–*ERG* fusion, can rewire multiple downstream processes, often amplifying additional oncogenic hits, such as the loss of *PTEN*. Understanding this intricate crosstalk highlights how AR signaling interacts with transcription factors to shape the gene networks that are fundamental to cancer progression. Furthermore, insights into how *TMPRSS2*–*ERG* integrates into broader molecular signatures offer the potential to refine patient stratification beyond traditional clinical parameters, emphasizing its importance as a biomarker and therapeutic target [55].

#### 2.3.5. Comparisons, Divergent Findings, and Future Outlook

While some groups have linked *TMPRSS2*–*ERG* positivity to unfavorable outcomes, others have observed no significant prognostic correlations. These discrepancies likely stem from differences in assay methodologies, such as fluorescence in situ hybridization, polymerase chain reaction (PCR), and RNA sequencing, which can influence detection sensitivity. Additionally, patient selection bias and coexisting genetic or epigenetic alterations that modify the impact of *ERG* fusions further contribute to inconsistent findings [56].

These advancements exemplify how *TMPRSS2*–*ERG* hijacks AR-responsive elements and often coexists with other key aberrations, such as the loss of *PTEN*. However, unresolved questions remain regarding the precise links between *ERG* fusion status and disease aggressiveness, as well as the optimal molecular context for targeting *ERG* or its co-regulators. These ongoing investigations are critical to refine our understanding of the functional effect of *ERG* on the pathophysiology of PCa [55].

Ultimately, *TMPRSS2*–*ERG* fusions represent a central node in the molecular architecture of PCa, demonstrating how transcription factors hijack androgen-responsive promoters to enhance oncogenesis [50]. Historically recognized for their prevalence, these fusions have now been found to be integrated into other pathways that govern genomic stability, cell motility, and survival. However, variability in clinical outcomes and methodological constraints underscore the need for robust integrative molecular analyses. Ultimately, clarifying the full impact of *ERG* on the pathophysiology of PCa remains an active area of investigation [55].

### 2.4. PTEN Loss, PI3K/AKT/mTOR Hyperactivation, and Crosstalk with AR 

#### 2.4.1. Historical to Current Perspective

*PTEN*, one of the most frequently inactivated tumor suppressors in PCa, initially gained attention because of its role in the negative regulation of cell survival pathways. Early studies on genetically engineered mouse models confirmed that the loss of *PTEN* accelerates tumorigenesis in the prostate, solidifying its importance in preserving normal epithelial homeostasis.

There is accumulating clinical data linking *PTEN* deficiency to high-grade tumors and aggressive disease phenotypes. Nevertheless, the broader landscape of PCa genomics has shown that *PTEN* status often intertwines with additional mutations, such as those in *TP53*, making straightforward prognostic or mechanistic conclusions more complex. Historically, *PTEN* has been recognized as a central gatekeeper that inhibits oncogenic cell growth. Modern genomic screening techniques, however, underscore not only the high frequency of *PTEN* deletions, but their synergy with other driver lesions, emphasizing the interconnected nature of genetic alterations in PCa [57].

#### 2.4.2. Mechanistic Insights

*PTEN* dephosphorylates phosphatidylinositol (3,4,5)-trisphosphate (PIP3), thereby limiting AKT activation. When *PTEN* is deficient, *PI3K/AKT/mTOR* signaling becomes hyperactive, fueling uncontrolled cell growth, metabolism, and survival. In PCa, this cascade may facilitate adaptation to low-androgen conditions by offering alternative survival routes [58].

*PTEN*-deficient tumors often exhibit compensatory modulation of the AR signaling pathway. When AR signaling is suppressed, PI3K/AKT activity increases, and vice versa, indicating a bidirectional feedback loop. This interplay exemplifies the convergence of multiple pathways in PCa, which reinforce each other when therapeutically inhibited. Hyperactivation of *PI3K/AKT/mTOR* can shift metabolic profiles to support rapid proliferation, while compensatory loops between the AR and *PI3K/AKT/mTOR* pathways demonstrate tandem behavior, where inhibition of one pathway frequently leads to the upregulation of the other [59]. The intricate relationship between AR signaling and the *PI3K/AKT/mTOR* pathway in advanced PCa is depicted in Figure 4. Loss of *PTEN* function, for instance, drives AKT hyperactivation, which can become more pronounced under conditions of AR inhibition, underscoring the need for combination therapeutic strategies.

#### 2.4.3. Methodological Constraints

Much of our knowledge of *PTEN* deficiency stems from cell lines or xenografts with distinct *PTEN* statuses. Although informative, these models lack the full intrapatient heterogeneity of diseases. For instance, they may not capture the dynamic interplay between *PTEN* loss, AR signaling, and additional oncogenic hits across different prostate tumor subclones.

Single-cell sequencing and patient-derived organoids have the potential to dissect *PTEN*-mediated molecular events in authentic tumor settings. However, these platforms are still evolving and variations in culture conditions or sequencing depths can limit their reproducibility or obscure rare subpopulations. Despite the robust evidence emerging from specific experimental systems, these findings may not fully capture the clinical spectrum of PCa. Integrating transcriptomics, proteomics, and epigenomics may provide a more comprehensive understanding of the nuanced roles of the loss of *PTEN* in different TMEs [60].

#### 2.4.4. Clinical or Scientific Significance

From a molecular pathophysiological standpoint, *PTEN* deficiency is the cornerstone for understanding how PCa cells escape normal growth restraints, as patients with *PTEN*-deficient tumors often exhibit more aggressive histopathological features—although co-mutations, such as those involving *TP53*, can sometimes overshadow the effect of loss of *PTEN* [61]. Moreover, the absence of *PTEN* demonstrates how dysregulation of a single node within the *PI3K/AKT/mTOR* pathway can orchestrate wide-ranging changes in cell survival and AR feedback loops, highlighting the intricate complexity of PCa signaling networks [62]. These findings show that aberrations in a single tumor suppressor can reshape multiple survival and proliferative pathways, while also extending their broader relevance to other malignancies in which *PTEN* is similarly compromised.

#### 2.4.5. Contrasting Evidence and Future Directions

While some studies have found strong associations between loss of *PTEN* and a poor prognosis, others have suggested that additional genetic alterations, such as *TP53* or *RB1* inactivation, may have a more dominant influence on outcomes. Consequently, *PTEN* status alone does not always provide clear prognostic discrimination.

*PTEN* deletion fundamentally disrupts cell cycle control and fosters the hyperactivation of the *PI3K/AKT/mTOR* pathway, illustrating its critical role in PCa pathogenesis [63]. However, open questions remain regarding how *PTEN* intersects with other frequently mutated pathways, and whether *PTEN* status alone reliably predicts clinical behavior. Historically viewed as a gatekeeper for normal cell cycle regulation, *PTEN* is now recognized for its broader influence on processes such as metabolic reprogramming and AR pathway crosstalk. The presence of co-mutations and compensatory signaling loops complicate the unidimensional view of *PTEN* deficiency. Further elucidation of the role of *PTEN* within the broader oncogenic network is pivotal for advancing our understanding of the pathogenesis of PCa.

### 2.5. Tumor Heterogeneity, Clonal Evolution, and Lineage Plasticity

#### 2.5.1. Historical to Current Understanding

PCa exhibits profound heterogeneity with variations among metastatic lesions in the same patient that evolve over time [64]. Early explanations of the progression of PCa focused on single “driver mutations” but failed to encompass the disease’s wide range of clinical behaviors. The advent of multiregional sequencing has revealed that individual metastatic sites often harbor distinct genetic alterations, challenging the notion of a uniform clonal driver [65]. Recent single-cell analyses have emphasized the dynamic nature of clonal populations as they adapt to selective pressures, highlighting the interplay between genetic diversity and environmental factors.

#### 2.5.2. Mechanistic Insights into Clonal Dynamics

Key molecular drivers, such as persistent AR signaling, DDR defects, and the loss of *PTEN*, independently confer survival benefits in different therapeutic contexts [61,63]. Over time, subclones harboring AR splice variants or traits of treatment-induced neuroendocrine PCa (t-NEPC) can outcompete other populations, resulting in a shifting hierarchy of dominant clones shaped by specific environmental or pharmacological pressures [66]. AR signaling remains a cornerstone of PCa biology; however, subclones can adapt through ligand-independent AR variants [65]. Additionally, alterations in the DDR or *PI3K/AKT/mTOR* pathways may emerge or expand under therapy, reinforcing resistant phenotypes. Some cells exhibit lineage plasticity, adopting t-NEPC characteristics, and transitioning to a small cell-like phenotype to evade AR-targeted approaches [66]. The evolution of PCa under therapeutic pressure can result in diverse subclonal populations, some of which acquire neuroendocrine features. Figure 5 illustrates this progression from an AR-driven phenotype to treatment-emergent neuroendocrine PCa, highlighting key genetic events, such as the loss of *RB1* and *TP53*.

#### 2.5.3. Methodological Constraints

Monitoring clonal evolution in real-time poses significant challenges because of the inherent limitations of current methodologies. Traditional biopsies are often invasive and can be technically and ethically difficult to repeat, particularly in metastatic settings. Circulating tumor DNA (ctDNA) offers a less-invasive alternative, providing a window for genomic alterations; however, rigorous validation is required to ensure its reliability and sensitivity [67]. Many studies have relied on single time points, which makes it difficult to track dynamic shifts in the subclonal architecture [68]. Although high-throughput sequencing has significantly improved our understanding, real-time longitudinal insights remain limited [64]. Moreover, single snapshots of the disease may fail to capture emerging or minority clones that eventually dominate therapy, introducing potential biases into the data [69].

#### 2.5.4. Clinical or Scientific Significance

Acknowledging tumor heterogeneity and clonal evolution is pivotal for understanding the molecular pathophysiology of PCa. Heterogeneity leads to varied responses to the same therapy, necessitating individualized treatment strategies [69]. Subclonal expansion often aligns with genetic or epigenetic changes, such as AR splice variants or neuroendocrine differentiation, enabling therapy escape. Furthermore, the shift to t-NEPC exemplifies how some prostate tumors can adopt a small cell-like phenotype, resembling other neuroendocrine malignancies both molecularly and histologically, with significant implications for prognosis and therapy design [70]. Recognizing phenotypic shifts improves risk stratification and illustrates how tumor cells adaptively rewire their signaling pathways under therapeutic stress, shedding light on their intricate biology [71].

#### 2.5.5. Comparisons and Divergent Data

Discrepancies often arise regarding the initiating factors of t-NEPC or other aggressive phenotypes. Some researchers emphasize *RB1* or *TP53* inactivation as essential early events, while others highlight the role of methylation patterns and chromatin remodeling, which may subtly shape the transition [70,71]. These contrasting perspectives underscore the need for harmonized study designs that incorporate both genetic and epigenetic profiling. Resolving whether specific mutations universally precede neuroendocrine transformation or broader epigenetic reprogramming drives this shift remains a critical question [70]. Despite these differences, there is agreement on the importance of the synergy between genomic lesions and epigenetic states. However, debate continues over the relative contribution of each factor and the precise sequence of events in clonal shifts [72].

#### 2.5.6. Future Outlook

Single-cell omics and longitudinal ctDNA analyses provide higher-resolution tracking of subclonal populations, and when combined with advanced imaging techniques—such as radiomic approaches that capture phenotypic diversity—these strategies illuminate the real-time evolution of clones [67]. Furthermore, pattern recognition and predictive modeling can forecast the emergence of aggressive subclones prior to overt clinical progression, thereby offering an earlier window for intervention [64]. Collectively, tumor heterogeneity, clonal evolution, and lineage plasticity underscore the complexity of prostate cancer, as historical multiregion studies have dismantled the simplistic notion of a single-driver event, and single-cell analyses have reinforced that evolving subclones exploit distinct molecular pathways—from androgen receptor variants to transitions toward treatment-emergent neuroendocrine prostate cancer—to overcome therapeutic barriers [68]. Future efforts that harness real-time genomic monitoring and high-resolution single-cell methods will help clarify the precise underpinnings of these adaptive processes, ultimately guiding more nuanced disease stratification and potentially intercepting lethal disease progression at earlier stages [73].

### 2.6. The Tumor Microenvironment and Immune Dynamics

#### 2.6.1. Historical to Current Understanding

The TME in advanced PCa comprises a complex network of cancer cells, stromal elements, immune populations, and secretory factors. Historically, most studies have focused on stromal fibroblasts and angiogenesis. More nuanced immunological studies have revealed a spectrum of tumor-associated immune cells, including myeloid-derived suppressor cells (MDSCs), TAMs, and regulatory T cells (Tregs), which collectively impair cytotoxic T cell activity. These findings have transformed the view of PCa from a purely “cold” tumor to one adept at deploying multiple immunosuppressive strategies, emphasizing the dynamic interplay of stromal and immune elements in shaping disease progression [74,75].

#### 2.6.2. Mechanistic Insights

PCa cells secrete key immunosuppressive cytokines, such as TGF-β and IL-10, and frequently express immune checkpoints, like PD-L1, collectively dampening T cell–mediated responses [75,76]. In addition, MDSCs and TAMs infiltrate the TME, releasing further factors that stifle cytotoxic lymphocytes, and thereby preserve a pro-tumor milieu [77]. In the context of bone metastasis, osteoblastic lesions disrupt normal bone remodeling, creating a niche that compromises immune surveillance [78]. This unique interaction between tumor cells and the bone environment fosters local immunosuppression, enabling metastatic lesions to evade host defenses. Moreover, interactions between PCa cells and the surrounding stromal and immune compartments further influence disease progression, as tumors adaptively reconfigure the cytokine landscape and immune checkpoint expression in response to selective pressure, highlighting the dynamic nature of immune evasion mechanisms [69]. Although PCa has historically been categorized as an immunologically “cold” tumor, multiple immunosuppressive factors and cells orchestrate a complex TME. Figure 6 outlines the spatial distribution of key immune cells, such as MDSCs and Tregs, and emphasizes how they inhibit antitumor responses.

#### 2.6.3. Methodological Constraints

Investigating the interplay between PCa cells and the immune system is a significant challenge. Standard two-dimensional cell cultures fail to replicate the complexity of the bone microenvironment and its immune components [78]. While xenograft and organoid systems capture some tumor–stroma interactions, they lack an intact human immune system [79]. Furthermore, assessing the dynamic changes in immune cell populations over time remains difficult without repeated tissue or liquid biopsies.

Insights from patient-derived xenografts and organoids provide partial authenticity, but still fail to reflect real-time immune dynamics. The complexity of the immune landscape also makes it challenging to distinguish which immunosuppressive elements are primary drivers versus passive bystanders, necessitating more sophisticated preclinical models and longitudinal studies in patients [80].

**Figure 6 biomolecules-15-00625-f006:**
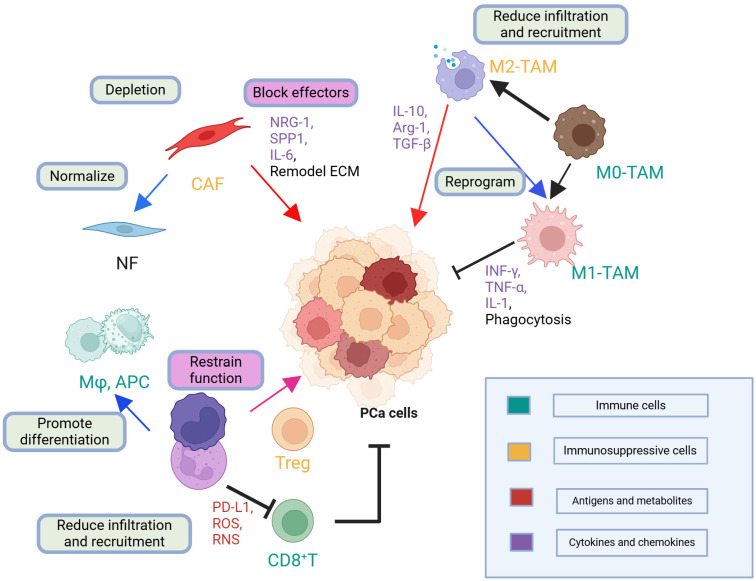
Immune Landscape and TME in Prostate Cancer. This schematic highlights the complex interactions among PCa cells, immune cells, and stromal components within the TME. NFs help maintain tissue homeostasis, whereas CAFs secrete factors such as NRG-1, *SPP1*, and IL-6 that remodel the ECM and support tumor progression. TAMs originate from M0 precursors and can polarize into M1 or M2 phenotypes: M1-TAMs secrete pro-inflammatory cytokines (e.g., IFN-γ, TNF-α, IL-1) that facilitate antitumor immunity; whereas, M2-TAMs produce immunosuppressive mediators (e.g., IL-10, Arg-1, TGF-β), enhancing tumor growth and immune evasion. Tregs further suppress antitumor responses by inhibiting CD8^+^ T cell activity through mechanisms involving PD-L1, ROS, and other immunosuppressive factors. Collectively, these dynamic cellular and molecular interactions shape the immune milieu in prostate cancer, driving disease progression and influencing therapeutic responses. Abbreviation: APC, antigen-presenting cell; Arg-1, arginase-1; CAF, cancer-associated fibroblast; CD8^+^ T, CD8^+^ T cell; ECM, extracellular matrix; IFN-γ, interferon-gamma; IL-1, interleukin-1; IL-10, interleukin-10; M0-TAM, M0 tumor-associated macrophage; M1-TAM, M1 tumor-associated macrophage; M2-TAM, M2 tumor-associated macrophage; Mφ, macrophage; NF, normal fibroblast; NRG-1, neuregulin-1; PCa, Prostate Cancer; PD-L1, programmed death-ligand 1; RNS, reactive nitrogen species; ROS, reactive oxygen species; *SPP1*, secreted phosphoprotein 1; TGF-β, transforming growth factor-beta; TNF-α, tumor necrosis factor-alpha; Treg, regulatory T cell. Created with Biorender (Accessed on 1 April 2025).

#### 2.6.4. Clinical or Scientific Significance

From a molecular pathophysiological perspective, understanding the immune interplay within the prostate TME is crucial. Immune-related biomarkers can help to stratify patients more accurately and identify those who may respond better to immunomodulatory interventions [81]. Evidence also suggests that DDR defects or shifts in AR signaling may influence immune cell recruitment and functionality, revealing a complex interaction between oncogenic pathways and the immune system [82].

Clarifying the molecular mediators of immune escape can help to elucidate how prostate tumors resist or remain dormant under immune pressure. Additionally, the TME reflects an intricate network of signals from AR pathways to the PI3K/AKT pathway, which collectively shapes tumor-immune dynamics and underscores the potential for integrated therapeutic strategies. PCa progression and therapeutic resistance are significantly influenced by the surrounding TME, which includes a variety of stromal and immune components [83]. Table 2 summarizes the key TME players, their secreted factors, and the mechanisms by which they contribute to immune suppression and cancer progression.

#### 2.6.5. Comparisons and Divergent Data

Clinical trials evaluating immunotherapeutic approaches for PCa have yielded mixed results. Some studies have highlighted the advantages of blocking MDSC recruitment or combining immunotherapy with DDR-targeted strategies, which partially improve immune responsiveness [84]. However, other trials have reported limited response rates, often attributing these modest outcomes to heterogeneous patient populations, variations in immune profiling methods, and the use of endpoints that fail to capture delayed immunological benefits [85].

The TME of PCa is broadly recognized as immunosuppressive, suggesting that successful therapeutic strategies require multi-faceted approaches. Despite this agreement, debates persist regarding which immune cell types—MDSCs, Tregs, or TAMs—present the most viable targets and how best to evaluate their clinical activity.

#### 2.6.6. Future Outlook

Advancements in single-cell immunophenotyping and high-resolution imaging provide deeper insights into the composition and spatial organization of immune cells within the TME. These insights indicate that the prostate TME is neither as inert nor as straightforward as previously thought; rather, a combination of immunosuppressive cytokines, regulatory immune cells, and a unique bone niche orchestrates an environment that shelters cancer cells from robust immune clearance. While existing models offer crucial clues, they remain imperfect at recapitulating the full complexity of human disease, and future advances in in situ single-cell techniques, next-generation three-dimensional organotypic systems, and integrative analyses are set to increase our understanding of immune tumor crosstalk, ultimately facilitating better molecular stratification and improved therapeutic approaches tailored to the immune landscape of PCa [80,86].

**Table 2 biomolecules-15-00625-t002:** Tumor Microenvironment (TME) Components and Immune Evasion Mechanisms in Prostate Cancer.

TME Component	Key Secreted Factors/Regulatory Signals	Main Function/Immune Evasion Mechanism	Clinical Implications
TAMs	IL-10, TGF-β, PD-L1	Suppress cytotoxic T-cell responses, promote angiogenesis and tumor growth	Elevated TAMs infiltration correlates with poor prognosis; potential therapeutic target for immunomodulation
MDSCs	Arginase, ROS, NO, IL-10	Inhibit T-cell function and antigen presentation, reinforce immunosuppression	High MDSCs levels linked to rapid disease progression; blocking MDSC recruitment or function may enhance immunotherapy outcomes
Tregs	IL-10, TGF-β	Suppress effector T-cell activity through cytokine-mediated and cell-contact mechanisms	Increased Tregs infiltration is associated with worse prognosis; Treg depletion or functional blockade may boost response to immunotherapy
CAFs	TGF-β, FGF, CXCL12, extracellular matrix components	Provide structural support, secrete growth factors that promote tumor invasion, and create immunosuppressive niches	FAP has emerged as a potential target; CAFs remodeling could improve T-cell infiltration in solid tumors
Bone Metastasis Niche (Osteoblasts/Osteoclasts)	RANKL, OPG, TGF-β	Remodel bone microenvironment to favor tumor cell growth, contribute to immune evasion through altered cytokine milieu	Central to skeletal metastases in prostate cancer; targeted RANKL inhibition (e.g., denosumab) or other bone-directed therapies may impair tumor progression
Endothelial Cells & Angiogenic Factors	VEGF, FGF, PDGF	Induce neovascularization that sustains tumor growth and metastasis	Anti-angiogenic strategies may enhance immunotherapy by reprogramming the tumor vasculature and improving immune cell infiltration

Abbreviations: CAFs, Cancer-Associated Fibroblasts; CXCL12, C-X-C Motif Chemokine Ligand 12; FAP, Fibroblast Activation Protein; FGF, Fibroblast Growth Factor; IL-10, Interleukin-10; MDSCs, Myeloid-Derived Suppressor Cells; NO, Nitric Oxide; OPG, Osteoprotegerin; PDGF, Platelet-Derived Growth Factor; PD-L1, Programmed Death-Ligand 1; RANKL, Receptor Activator of Nuclear Factor Kappa-B Ligand; ROS, Reactive Oxygen Species; TAMs, Tumor-Associated Macrophages; TGF-β, Transforming Growth Factor-beta; TME, Tumor Microenvironment; Tregs, Regulatory T Cells; VEGF, Vascular Endothelial Growth Factor.

## 3. Molecular Stratification and Diagnostic Advances

Technological breakthroughs in genomic, transcriptomic, and epigenomic profiling, combined with ever-evolving imaging modalities, have transformed the classification of mPCa. Historically, simpler diagnostic assays have offered limited insight into the molecular underpinnings of the disease; however, contemporary approaches allow for more refined stratification, enabling clinicians to match patients with targeted therapies against specific molecular aberrations [87]. Despite these advances, significant challenges remain, including limited global access, cost constraints, and varying guidelines for the implementation of advanced diagnostics [88].

A variety of genomic and immunohistochemical assays have been adopted in clinical practice to identify actionable alterations in PCa, from *BRCA2* and *ATM* mutations to AR-*V7* status [89]. Table 3 summarizes the key diagnostic panels and recommended biomarkers, highlighting their respective clinical utilities and current limitations.

### 3.1. High-Resolution Molecular Profiling

From a historical perspective, the introduction of targeted next-generation sequencing (NGS) panels—typically covering 50 to 500 genes—has revolutionized cancer diagnostics by enabling rapid, cost-effective detection of actionable driver mutations, such as those in *BRCA2* or *PTEN* [90]. As these panels have become increasingly comprehensive, the field has evolved to incorporate whole-exome and whole-genome sequencing (WES/WGS) approaches, which have not only facilitated the identification of complex structural rearrangements and rare genetic variants, but have signaled a paradigm shift from broad histopathological classification to precision oncology driven by molecular signatures [91].

In current clinical practice, many institutions have adopted a tiered testing strategy that begins with focused gene panels and escalates to WES/WGS when the initial results are ambiguous or when a patient’s clinical risk profile justifies a more in-depth inquiry [92]. While targeted NGS remains more accessible and directly actionable, WES/WGS often uncovers potentially meaningful, but less common, alterations, although the additional complexity can present interpretation challenges that are frequently addressed by cross-disciplinary molecular tumor boards [93]. Although some centers report high concordance between gene panels and broader sequencing approaches for common mutations, others note that extended analyses reveal rare variants that may influence therapeutic decision-making—discrepancies that likely stem from differences in sequencing depth, analytical pipelines, and patient selection criteria [94].

Looking to the future, as costs continue to decline and AI-driven analytics become more integrated, these next-generation techniques are expected to play an even more central role, although the extent to which WES/WGS will be routinely applied, especially in resource-limited settings, remains to be seen [95]. Complementing DNA-based approaches with transcriptomics via RNA sequencing has enhanced our understanding by identifying functionally active genes and alternative splicing events, such as AR-*V7* [96], while epigenomic assays, such as assay for transposase-accessible chromatin using sequencing and chromatin immunoprecipitation-sequencing, have revealed regulatory elements and super-enhancers that may drive therapeutic resistance [97]. At the frontier of molecular profiling, single-cell sequencing captures subclonal diversity with unprecedented resolution, offering the promise of early detection of therapy resistance and dynamic treatment adaptation [98]. Nevertheless, the interpretation of epigenetic and transcriptomic data is often context-dependent and requires functional validation, with variability in tissue acquisition and RNA quality complicating cross-study comparisons [99]. For these methods to influence clinical decisions effectively, robust frameworks for data interpretation and standardized reporting are essential—an area that continues to evolve [100]. Despite the high cost and complexity that currently limit the routine use of single-cell sequencing, its potential to unveil emergent resistant clones in near-real-time underscores its promise for the future of precision oncology [101].

### 3.2. Companion Diagnostics and Gene Panels

#### 3.2.1. DDR-Focused Panels for PARP Inhibitor Selection

The discovery of *BRCA1/2* mutations as potent drivers of tumorigenesis revolutionized breast and ovarian cancer treatment, and prostate cancer soon followed suit.

Today, commercial assays commonly include those for *BRCA1*, *BRCA2*, *ATM*, and other DDR-related genes, reshaping diagnostic paradigms by enabling clinicians to identify patients who may benefit from PARP inhibitor treatment.

The U.S. Food and Drug Administration’s approval of these agents for mCRPC has further underscored the necessity of systematic genetic testing [102]. Germline mutations in DDR-related genes not only impact patients but have implications for their relatives, emphasizing the ethical and practical importance of genetic counseling [103]. However, many patients still lack access to genetic testing because of its cost or geographical barriers, and the predictive value of less-common mutations in DDR-related genes remains underexplored, highlighting a significant gap in the existing evidence [102]. By identifying DDR deficiencies, clinicians can more accurately administer PARP inhibitors—a prime example of stratified medicine. However, the emergence of reversion mutations and evolving resistance patterns cautions against viewing these biomarkers as static or universally reliable [103].

#### 3.2.2. AR Variant Detection and *PTEN*/PI3K Panels

In addition to tests for DDR-related genes, those for AR splice variants, such as AR-*V7*, and for LBD mutations are proving to be useful at predicting responses to second-generation AR inhibitors [104]. Parallel testing of *PTEN* status and *PIK3CA*/AKT mutations, often conducted through immunohistochemistry or targeted sequencing, has been used to further refine patient eligibility determinations for clinical trials evaluating *PI3K*/AKT inhibitors [63]. Despite challenges with reimbursement and varying guideline structures across regions, the shift toward precision oncology continues to drive the clinical adoption of these gene panels [105]. However, divergent evidence exists. Some studies have shown a strong correlation between AR variants and a poor response to AR-targeted therapies, while others have highlighted the role of co-occurring alterations, such as those in *TP53*, in influencing therapeutic resistance [106]. This variability underscores the multifactorial nature of treatment resistance. Looking forward, AI-powered predictive models and CRISPR-based functional assays hold promise for deepening our understanding of how alterations in AR and *PTEN* intersect to affect therapeutic outcomes, potentially guiding more personalized treatment strategies [107].

#### 3.2.3. Performance Metrics of Key Molecular Biomarkers

Despite the clear clinical utility of testing for biomarkers, such as AR-*V7*, DDR-related gene mutations, and the loss of *PTEN*, the sensitivity and specificity of these assays vary widely across studies. Table 4 summarizes the published ranges of sensitivity, specificity, and positive/negative predictive values for several commonly assessed molecular biomarkers in PCa. These estimates should be interpreted with caution because different assays (e.g., immunohistochemistry, PCR-based methods, NGS) and patient populations can yield heterogeneous results.

### 3.3. Advanced Imaging: PSMA PET-CT and Beyond

Conventional imaging techniques, such as CT and bone scans, often underestimate metastatic lesions because of their limited resolution, but the introduction of PSMA PET-CT has made it possible to detect subclinical or oligometastatic disease with far greater accuracy, thus enabling more precise therapeutic interventions [110]. This breakthrough has proven especially valuable for patients with biochemical recurrence, as it informs localized salvage treatments and radioligand strategies [111]. Despite its high sensitivity, however, PSMA PET-CT can be less effective for identifying neuroendocrine or small-cell variants, which typically exhibit low levels of or absent PSMA expression—an important limitation when every lesion, no matter how small, should be considered for tailored management [112]. Moreover, medical centers without PSMA PET-CT technology risk compromising their ability to detect micro-metastases and provide equitable care, raising concerns about the accessibility gap [113].

To circumvent this shortcoming, multiple Technetium-99m (99mTc)-labeled PSMA ligands have been developed for use with single photon emission computed tomography (SPECT)/CT, which is more widely available than PET-CT in many regions. The findings of a recent meta-analysis support the diagnostic potential of 99mTc-based SPECT/CT tracers for detecting PCa lesions, suggesting that they may offer a more accessible, cost-effective alternative for centers lacking PET-CT infrastructure [114].

Meanwhile, experimental tracers targeting integrins or fibroblast activation proteins (FAPs) are under active investigation, reflecting the growing interest in visualizing multiple components of the TME [115]. Radiogenomics—the integration of imaging data and genomic profiles—further expands this landscape, offering the potential to monitor tumor biology in real-time, without invasive procedures [116]. However, challenges persist, including the lack of standardized imaging protocols, limited tracer availability, and inconsistent reporting metrics [117]. The incorporation of AI and machine learning may help interpret complex radiogenomic datasets, potentially detecting early patterns of disease progression [118]. Nevertheless, cost and regulatory barriers continue to limit widespread adoption, underscoring the need to balance innovation with real-world feasibility. These developments highlight the evolution of PCa diagnostics from rudimentary classification to increasingly sophisticated molecular and imaging-based approaches, which have already revolutionized patient stratification and outcome prediction. Although discrepancies in reported findings—regarding AR variants, DDR-related gene mutations, or imaging modalities—reflect the inherent complexity of treatment resistance, there is growing consensus that molecular stratification is critical for precision care [119].

Looking ahead, emerging technologies, such as chimeric antigen receptor (CAR)-T cell therapy, show potential for highly specific antigen targeting, but on-target/off-tumor toxicity and immunosuppressive microenvironments remain formidable obstacles [120]. For example, to correct germline *BRCA2* mutations, CRISPR-based gene editing also holds promise, but ensuring both safety and target specificity is paramount [121]. AI-driven analytics may eventually be used to integrate genomic, imaging, and clinical data on a large scale to uncover novel insights, provided that the quality control of input data remains rigorous [122]. Overall, molecular stratification and innovative diagnostics are transforming the management of metastatic PCa by refining disease classification, guiding individualized treatment choices, and detecting resistance at earlier stages, even as affordability, standardization, and regulatory issues persist [87]. However, the synergy of multi-omic profiling, cutting-edge imaging, and computational tools points toward a future where genuine patient-tailored therapy becomes the standard of care.

### 3.4. Liquid Biopsies in Metastatic Prostate Cancer

#### 3.4.1. Historical Context and Technological Evolution

Liquid biopsy has emerged as a minimally invasive approach for capturing molecular information from mPCa, building on the earlier research on circulating tumor cells (CTCs) and cell-free DNA (cfDNA) from other solid tumors [123]. Initial efforts in the early 2000s demonstrated that enumerating CTCs could provide prognostic insights; however, subsequent technological refinements expanded the scope to include ctDNA, exosomes, and other blood-based analytes [124]. Liquid biopsy strategies have gained significant traction in the management of mPCa, offering minimally invasive insights into tumor evolution and therapeutic resistance. Table 5 highlights the key liquid biopsy modalities, their respective analytical methods, and current clinical applications, providing a comparative overview of the advantages and limitations associated with each approach. These assays offer a window into tumor heterogeneity and dynamic clonal evolution, circumventing the challenges of repeated tissue biopsies at metastatic sites [125].

#### 3.4.2. Methodological Approaches and Clinical Relevance

Advanced platforms can be used to isolate viable tumor cells from peripheral blood, enabling downstream analyses, such as immunophenotyping, genomic sequencing, and functional assays. Although CTC enumeration has been linked to survival outcomes, emerging techniques now probe gene expression, AR variants, and epithelial–mesenchymal transition markers to gain deeper insights into resistance mechanisms [126]. Meanwhile, NGS of ctDNA can detect actionable mutations (e.g., those in *BRCA2*, AR, and *PIK3CA*) and track tumor burden over time. Moreover, longitudinal sampling helps identify emerging resistance pathways—particularly under selective pressure from AR-targeted therapies or PARP inhibitors—allowing clinicians to adapt treatments before overt clinical progression occurs [127]. In addition, exosomes containing tumor-derived proteins, RNA, and DNA have garnered interest for their potential to reflect the real-time state of metastatic lesions. Although protocols for the isolation and characterization of exosomes remain less standardized, exosome-based assays may complement or refine ctDNA analysis [128].

#### 3.4.3. Critical Assessment and Significance

Liquid biopsies address many of the limitations of tissue biopsies, including sampling bias and procedural risks, by offering a noninvasive, real-time window into tumor biology. This approach is especially relevant in mPCa, where subclonal differences among bone, lymph nodes, and visceral lesions can be significant [89]. Clinical trials have begun incorporating ctDNA- or CTC-based endpoints to stratify patients for experimental therapies. For example, the detection of AR-*V7* in CTCs can inform decision-making on whether to continue with AR-targeted agents or switch to chemotherapeutic or other targeted strategies [129]. While some studies have demonstrated high concordance between mutations identified via liquid biopsy and those detected through tissue-based sequencing, others have revealed discrepancies that may arise from a low circulating tumor fraction or spatial heterogeneity, underscoring the need to harmonize assay sensitivity and specificity across different platforms [130].

#### 3.4.4. Methodological Challenges

Variations in pre-analytical processing, assay design, and bioinformatics pipelines can yield inconsistent results, highlighting the need for a standardized framework for ctDNA and CTC collection, storage, and analysis. Such a framework is still under development [131]. Detecting low-frequency variants, such as rare mutations in cfDNA, demands ultra-deep sequencing coupled with rigorous quality control measures to minimize false positives, which is critical for identifying subclonal populations that may drive treatment resistance [24]. Moreover, despite the less-invasive nature of liquid biopsies compared to tissue biopsies, high costs and barriers to insurance coverage continue to limit their widespread adoption, particularly in resource-limited settings [132].

#### 3.4.5. Future Outlook

Machine-learning algorithms may refine the interpretation of liquid biopsy data by correlating mutational patterns with imaging findings, clinical outcomes, and other biomarkers, potentially predicting impending relapse or identifying the optimal switch in therapy [133]. Combining cfDNA mutation analysis with transcriptomic or epigenomic signatures from exosomes may provide a more comprehensive understanding of tumor evolution [134]. Beyond the mere detection of mutations, next-generation platforms aim to culture CTCs ex vivo or use exosomal content to assess drug susceptibility and guide real-time therapeutic adjustments [135]. Thus, liquid biopsies represent a transformative diagnostic modality for mPCa, offering insights into tumor heterogeneity and resistance dynamics. As technology matures and clinical validation increases, blood-based assays may become integral to precision oncology, enabling agile and personalized treatment strategies [105]. The success of precision oncology depends on standardized methodologies, robust validation in prospective trials, and thoughtful integration into existing clinical workflows.

## 4. Targeted Therapeutic Approaches

As PCa management has evolved, the integration of next-generation agents and combination strategies has reshaped clinical practice. Historically, therapies have been limited to nonspecific hormonal suppression and cytotoxic regimens; however, modern approaches aim to exploit specific molecular vulnerabilities. This section outlines major targeted treatments, focusing on their clinical utility, evidence for their efficacy, and their prospects.

### 4.1. AR Axis–Centric Treatments

Enzalutamide, apalutamide, and darolutamide bind to the AR with high affinity, providing overall survival (OS) benefits in both hormone-sensitive and castration-resistant settings; while abiraterone, by blocking *CYP17A1*-mediated androgen biosynthesis, has shown similar efficacy in pivotal clinical trials [29]. Despite these advances, resistance remains inevitable in many patients, driven by AR mutations, AR-*V7* splice variants, and upregulated bypass pathways [136]. Early AR blockade therapies revolutionized treatment, but were limited by short-lived responses. By contrast, second-generation anti-androgen drugs and abiraterone have significantly extended survival in large randomized trials [137].

Bipolar androgen therapy (BAT), which involves the monthly administration of high-dose testosterone to patients already receiving ADT, has emerged as another intriguing approach to overcome resistance mechanisms. In the phase II TRANSFORMER trial, BAT demonstrated significant clinical efficacy in patients who had progressed on abiraterone, and notably, appeared to resensitize tumors to subsequent enzalutamide therapy [138]. In a more recent phase II study (COMBAT), BAT in combination with nivolumab was also efficacious. Moreover, metastatic tumor biopsies showed that BAT induced pro-inflammatory gene expression changes in patients achieving a clinical response, suggesting potential synergy with immunotherapy [139]. These findings highlight the paradoxical utility of high-dose testosterone treatment, which may disrupt AR-dependent survival circuits and augment the immune milieu within the TME.

Newer combination strategies are emerging, such as the use of PROTACs to harness the ubiquitin–proteasome system for degrading full-length AR and its variants, DBD inhibitors that circumvent resistance stemming from the LBD alterations, and epigenetic co-targeting approaches using LSD1 or bromodomain and extra-terminal (BET) inhibitors to modify AR-centric chromatin states, potentially delaying or preventing resistance [140]. In particular, PROTAC technology degrades target proteins via the ubiquitin–proteasome system, overcoming the limitations of traditional inhibitors by targeting “undruggable” proteins and reducing drug resistance [141]. In mCRPC, ARV-110 targets the AR, and has demonstrated a more than 50% reduction in prostate-specific antigen (PSA) levels in AR-mutant patients with good tolerability, while ARV-766 shows broader efficacy with lower toxicity [141]. Additionally, BET degraders, such as ARV-771, inhibit BRD4—a key transcription factor in CRPC progression—leading to robust tumor regression [142]. Despite ongoing challenges in optimizing drug delivery and managing dose-dependent hook effects, PROTAC technology represents a promising strategy for treating drug-resistant prostate cancer, with potential applications in other malignancies. However, many of the trials supporting these combination regimens involved limited patient numbers, which make it difficult to generalize toxicity profiles or long-term efficacy. Looking forward, ongoing efforts should include AI-based analyses of AR structural changes to predict resistance patterns and guide drug design, while the use of CRISPR technology may enable more precise modeling of AR mutations, thereby facilitating improved risk stratification and the development of personalized therapies [143,144].

### 4.2. DDR-Defect-Based Therapies: PARP Inhibitors and Beyond

Building on the concept that tumors with DDR deficiencies can be selectively killed, olaparib and rucaparib have gained approval for mCRPC with *BRCA* mutations or other alterations in DDR-related genes, and the PROfound trial demonstrated that *PARP* inhibition can outperform certain AR-targeted therapies in biomarker-selected populations, underscoring the clinical importance of systematic genetic testing [145]. These findings have spurred interest in synergistic combinations, where AR blockade may amplify DNA damage and thereby enhance the cytotoxicity of PARP inhibitors, while ATR or DNA-PK inhibitors also show promise in intensifying therapeutic responses, albeit with higher toxicity [8]. Although *BRCA2* mutations strongly predict benefit, the variable sensitivity of other DDR-related gene alterations highlights the need to optimize predictive biomarkers to prevent undue toxicity in non-responders [46]. Overall, PARP inhibitors represent a major step toward precision oncology, as their use has confirmed that targeted therapy can be highly effective in genomically defined subgroups. However, the emergence of acquired resistance—such as through reversion mutations—underscores the dynamic interplay between treatment and tumor evolution [146]. In parallel, several DDR-targeted therapy trials are ongoing for PCa, including a phase 2 trial of carboplatin (NCT03148795) in HRR-mutated mCRPC that began enrollment in September 2023, and focuses on reducing PSA levels and improving progression-free survival [147], and the GUNS trial (NCT04812366), which is an adaptive umbrella study of high-risk localized disease that includes niraparib for DDR-deficient tumors [148]. Furthermore, the EvoPAR-Prostate01 trial (NCT06120491) is investigating the combination of saruparib, a selective *PARP1* inhibitor, with an AR pathway inhibitor in metastatic hormone-sensitive prostate cancer (mHSPC), stratifying patients based on HRR mutation status [149]. Collectively, these efforts emphasize the ongoing pursuit of biomarker-driven and combination strategies to advance personalized therapies for PCa.

### 4.3. Targeting PI3K/AKT/mTOR and WNT Pathways

*PI3K/AKT/mTOR* hyperactivation frequently coincides with *PTEN* loss, rendering it a prime target for drug development. CAPItello-281 (NCT04493853) is an ongoing Phase 3, randomized, double-blind trial evaluating the efficacy and safety of capivasertib, a selective oral pan-AKT inhibitor, in combination with abiraterone and ADT in patients with de novo mHSPC and *PTEN* loss, a deficiency common in prostate cancer that leads to PI3K/AKT pathway hyperactivation and resistance to AR inhibitors. The trial aims to enroll 1012 patients, with radiographic progression-free survival (rPFS) as the primary endpoint and OS as a secondary endpoint. Preliminary results indicate that capivasertib significantly improves rPFS compared to placebo, with a trend toward OS benefit, and the safety profile aligns with prior data; final results are expected by 2027 [150]. Clinical trials with AKT inhibitors, such as ipatasertib and capivasertib, when combined with AR blockade, have demonstrated modest benefits, particularly in *PTEN*-deficient prostate cancers; however, adverse effects like hyperglycemia and dermatological toxicities limit the therapeutic window [151]. Emerging evidence implicates WNT pathway dysregulation in disease progression and neuroendocrine transdifferentiation under potent AR suppression, with preclinical data suggesting that inhibiting WNT ligand secretion or β-catenin may help prevent treatment-emergent neuroendocrine prostate cancer [152]. Additionally, epigenetic modulation through agents such as LSD1 or *EZH2* inhibitors appears to preserve epithelial differentiation and delay the onset of aggressive phenotypes [153]. Although synergistic effects of multi-pathway inhibition (e.g., combining AR, AKT, and WNT targeting) have been reported, the scarcity of late-phase trials and the challenge of overlapping toxicities underscore the need for level I evidence from robust randomized studies.

### 4.4. Immuno-Oncology Approaches

Despite the transformative impact of checkpoint inhibitors in melanoma and lung cancer, prostate cancer has seen limited success, with benefits largely confined to select molecular subsets, such as mismatch repair-deficient or *CDK12*-mutant tumors [154]. Experimental strategies—such as combining checkpoint blockade with radiation or tumor vaccines—aim to improve immune infiltration and overcome the immunologically “cold” tumor microenvironment [155]. In parallel, cellular therapies like CAR-T cells targeting PSMA or *PSCA* hold promise, although their potential is currently hindered by T cell exhaustion and “on-target, off-tumor” toxicities; novel approaches that engineer T cells with dominant-negative PD-1 or IL-12 expression are being explored to enhance durability [156]. Additionally, efforts in myeloid reprogramming—such as blocking angiogenesis or targeting FAP—seek to reshape the stromal environment to facilitate more effective T cell responses [157]. These advances are particularly important because immunotherapy offers the potential for durable control in patients who fail conventional AR- or DDR-targeted treatments. However, comprehensive prospective trials are essential to establish optimal combinations, address toxicity management, and identify robust predictive biomarkers [158].

Building on this evolving immunotherapy landscape, six-transmembrane epithelial antigen of prostate 1 (*STEAP1*) has emerged as a promising immunotherapeutic target for mCRPC because of its selective tumor expression and accessible cell-surface localization [156]. Bispecific T cell engagers (BiTEs), such as xaluritamig (AMG 509), simultaneously bind *STEAP1* on cancer cells and CD3 on T cells, bridging them into close proximity, resulting in direct and potent activation of cytotoxic T cells to induce tumor cell killing [159]. In a recent phase I trial (NCT04221542), xaluritamig demonstrated significant reductions in PSA levels, partial clinical responses, and durable benefits exceeding 6 months in high-dose cohorts. However, immune-related toxicities, notably cytokine release syndrome, fatigue, and anemia, constrained dosing to a weekly maximum tolerated dose of 1.5 mg [160]. Further enhancing therapeutic efficacy, the next-generation BiTE candidate BC261 has demonstrated superior tumor eradication in preclinical models, achieving approximately 30-fold greater T cell infiltration even against tumors with low *STEAP1* expression levels [159]. BiTE therapies targeting *STEAP1* offer advantages over CAR-T cells, including faster clinical availability; potent T cell activation, independent of prior immune priming; and combinational treatment potential. Nonetheless, overcoming immune-mediated toxicities and optimizing dosing regimens remain critical challenges that require additional clinical validation to integrate *STEAP1*-directed BiTE therapies fully into the prostate cancer treatment landscape [159].

### 4.5. PSMA-Targeted Radioligand Therapy

PSMA is overexpressed in most prostate cancers, making it an attractive target for radioligand therapy. The VISION trial demonstrated that adding ^177^Lu-PSMA-617 to standard care for heavily pretreated metastatic castration-resistant prostate cancer yields a survival advantage, reflecting a new treatment horizon for advanced disease [161]. Alternative alpha-emitting agents, such as ^225^Ac-PSMA-617, induce more potent DNA damage but are associated with significant salivary gland toxicity [162].

In addition to the potential for alpha emitters, beta emitters, such as terbium [^161^Tb], are also under investigation. A study of ^161^Tb-PSMA-I&T is currently being conducted in the phase I/II VIOLET trial [163]. Unlike ^177^Lu, ^161^Tb emits Auger and conversion electrons, which deposit a higher concentration of radiation over a shorter path, potentially offering superior tumor cell killing with reduced off-target toxicity. Preclinical data have shown greater in vitro and in vivo efficacy of ^161^Tb compared to ^177^Lu, although further clinical evaluation is warranted to establish optimal dosing, safety profiles, and long-term outcomes.

PSMA-targeted radioligand therapy delivers cytotoxic radiation directly to prostate cancer cells while sparing normal tissues. As shown in Figure 7, patients undergo PSMA PET-CT imaging to confirm target expression before receiving ^177^Lu-PSMA-617 or related alpha-emitting agents.

Several key trials are currently underway to evaluate PSMA-RLT for PCa. The CONVERGE-01 trial is investigating Ac-225 rosopatamab tetraxetan in PSMA-positive mCRPC to optimize dosing and evaluate efficacy, focusing on rPFS and OS. The PSMAction trial (phase 2/3) is assessing Ac-PSMA-617 in patients with mCRPC who have previously received Lu-177 therapy, with rPFS and OS as the primary endpoints [164]. The Satisfaction Trial (NCT05983198) is evaluating the safety and efficacy of Ac-PSMA-R2 in patients with Lu-177-pretreated and Lu-177-naïve PSMA-positive mCRPC [165]. Meanwhile, the EVOLUTION trial is exploring the combination of Lu-177-PSMA-617 with immune checkpoint inhibitors (ipilimumab + nivolumab) to enhance therapeutic synergy [166]. Finally, the ALPHABET trial is investigating Lu-177-PSMA-I&T in combination with radium-223 to determine the maximum tolerated dose and PSA response rates [167]. Collectively, these studies aim to improve the efficacy of PSMA-RLT by leveraging alpha-emitting isotopes and combination strategies, offering promising advancements in the treatment of mCRPC.

Patient selection for these therapies relies on PSMA-PET imaging to confirm target expression, and there is growing interest in combining PSMA-targeted radioligand therapy with PARP inhibitors, AR blockade, or immunotherapies to synergistically enhance tumor cell death, although overlapping toxicities remain a concern [168]. Moreover, most radioligand therapy studies have focused on end-stage disease, leaving it unclear whether earlier interventions would yield greater survival gains. Interim imaging or circulating biomarkers may help tailor doses and mitigate toxicity [169]. Radioligand therapy exemplifies precision radio-oncology by directly targeting tumor cells while sparing normal tissues, yet its widespread implementation is hindered by logistical challenges, cost, and the complexities of radiopharmaceutical handling [170].

Targeted therapies and combination regimens continue to redefine the management of mPCa, bridging historical insights into AR suppression with cutting-edge modalities, such as *PARP* inhibition, PI3K/AKT blockade, immunotherapy, and PSMA-targeted radioligand therapy [171]. Multiple targeted and immunotherapeutic approaches are under investigation or have recently demonstrated clinical benefit in advanced PCa, ranging from *PARP* inhibition in *BRCA1/2*-mutated disease to PSMA-targeted radioligand therapy in mCRPC [172]. Table 6 provides an overview of key trials and their outcomes, highlighting the progress and challenges associated with each strategy. While numerous trials have confirmed significant survival benefits, the heterogeneity across patient populations necessitates personalized strategies that are often guided by molecular diagnostics and advanced imaging techniques [173]. The field has evolved rapidly from basic hormonal manipulation to sophisticated targeted approaches, and disparate clinical trial outcomes often reflect differences in biomarker selection, trial design, and disease heterogeneity. There is a growing consensus that multi-omic profiling is vital for optimally matching treatments with tumor biology [174]. Ultimately, the therapeutic landscape of mPCa is dynamic and increasingly personalized, reflecting the convergence of robust clinical evidence, molecular diagnostics, and novel mechanistic insights. Long-term success will depend on equitable access, standardized protocols, and collaborative research aimed at overcoming current limitations and delivering durable patient benefits.

To provide a clear comparison of the principal treatment modalities for advanced PCa—including AR pathway inhibitors, PARP inhibitors, immuno-oncology strategies, and PSMA-targeted approaches—we summarize their key mechanisms, benefits, limitations, and associated biomarkers in Table 7.

## 5. Conclusions and Future Perspectives

In this review, we summarize a broad range of findings that underscore the molecular and clinical complexities driving mPCa. At the molecular level, key themes include the centrality of AR signaling, the importance of DDR pathways (particularly *BRCA1/2* and related genes), and the convergence of *PI3K/AKT/mTOR* and WNT signaling pathways. Studies of the TME have further highlighted the immunosuppressive barriers encountered in advanced disease. In clinical settings, newer AR inhibitors, PARP inhibitors, radioligand therapies, and emerging immuno-oncological approaches have all contributed to improved outcomes; however, resistance remains pervasive. Resistance is exacerbated by the heterogeneous and evolving genetic and epigenetic landscapes of metastatic lesions, underscoring the need for refined molecular stratification and real-time monitoring. Although genomic and imaging technologies have expanded substantially, obstacles such as cost, access, and lack of standardized interpretation limit their full-scale adoption in diverse clinical settings.

Conflicting evidence, particularly regarding biomarkers such as *TMPRSS2*–*ERG* fusions and AR splice variants, stems from methodological inconsistencies and highlights the necessity for harmonized protocols. By consolidating recent mechanistic discoveries and linking them to evolving clinical strategies, this review offers an integrated framework that spans the molecular, immunological, and therapeutic dimensions of mPCa. Clinically, the insights discussed—mapping core pathways, identifying known resistance mechanisms, and emphasizing the role of molecular diagnostics—provide strategic advantages for improving patient selection and optimizing therapeutic sequences.

Several opportunities and future agendas have emerged. First, synergistic clinical trials that systematically assess combination regimens (for instance, AR inhibition plus DDR-targeted or immunotherapeutic agents) may clarify the survival benefits and the molecular trajectories of resistance. Second, longitudinal biomarker tracking, particularly through liquid biopsies, may reveal early resistance pathways and enable adaptive therapy switching before overt clinical progression. Third, future trial designs may benefit from multi-omic, prospective cohorts that unify genomic, transcriptomic, and imaging data along with standardized clinical endpoints, potentially reconciling conflicting biomarker outcomes. The incorporation of advanced computational tools, including AI-driven analytics, will further integrate these data streams, while CRISPR-based functional assays may unearth new molecular targets within the AR signaling and DDR pathways. In parallel, more refined cellular therapies, such as CAR-T and natural killer cell platforms, may eventually overcome immunosuppressive obstacles in PCa if issues of specificity and toxicity can be resolved. Finally, global collaborative efforts coordinated by academic institutions, industry partners, and regulatory bodies are essential for validating assay standards, sharing biobanks, and rapidly translating bench discoveries into bedside interventions. Therefore, the field of mPCa is at a moment of convergence, and the integration of multi-omic insights, expanding therapeutic approaches, and computational advances has laid the groundwork for more precise and durable treatment strategies. Addressing the current evidence gaps, refining trial methodologies, and fostering cross-sector partnerships will be pivotal to ensure that these innovations yield tangible life-extending benefits for patients in real-world practice.

## Figures and Tables

**Figure 1 biomolecules-15-00625-f001:**
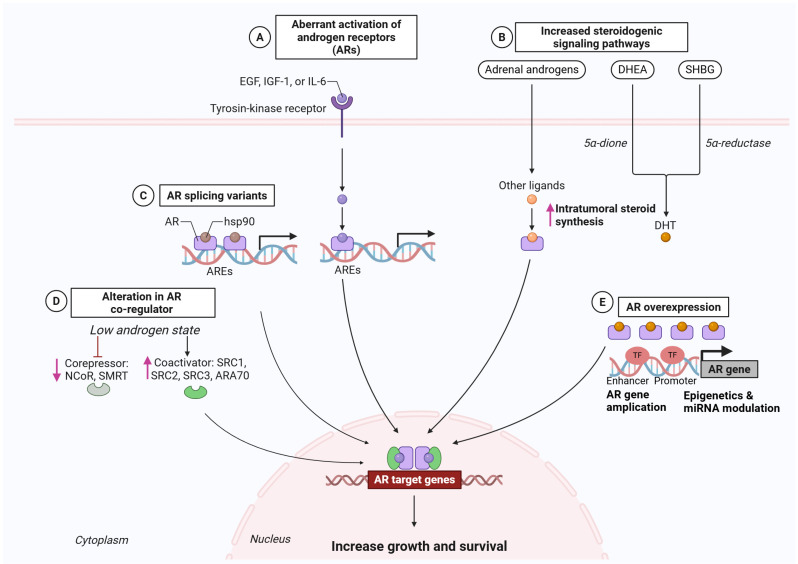
Mechanisms of AR Reactivation and Resistance. This schematic illustrates multiple pathways by which AR signaling can be reactivated in prostate cancer, ultimately driving therapy resistance. (A) Aberrant AR activation: Growth factors, such as EGF, IGF-1, or IL-6, can transactivate the AR via tyrosine-kinase receptors, enabling AR signaling even under low-androgen conditions. (B) Increased steroidogenic signaling pathways: Adrenal androgens and their precursors (e.g., DHEA) are converted intratumorally to DHT through 5α-reductase activity. Elevated levels of SHBG and other ligands also contribute to AR activation. (C) AR splicing variants: Certain variants (e.g., AR-*V7*) lack the ligand-binding domain, allowing constitutive AR target gene activation independently of androgen binding. (D) Alterations in AR co-regulators: Decreased expression of co-repressors (e.g., *NCoR* and *SMRT*) or increased expression of coactivators (e.g., SRC1, SRC2, SRC3, and *ARA70*) can further enhance AR-mediated transcription. (E) AR overexpression: AR gene amplification, epigenetic modifications, and miRNA dysregulation can lead to AR overexpression, thus amplifying AR signaling and promoting tumor growth. Together, these mechanisms converge to maintain or upregulate AR signaling despite therapeutic interventions, resulting in increased cancer cell survival and disease progression. Abbreviations: *ARA70*, androgen receptor coactivator 70; AR, androgen receptor; DHEA: dehydroepiandrosterone; DHT, dihydrotestosterone; EGF, epidermal growth factor; IGF-1, insulin-like growth factor-1; IL-6, interleukin-6; miRNA, microRNA; *NCoR*, nuclear receptor co-repressor; SHBG, sex hormone-binding globulin; *SMRT*, silencing mediator for retinoid or thyroid hormone receptor; *SRC*, steroid receptor coactivator. Created with Biorender (Accessed on 1 April 2025).

**Figure 2 biomolecules-15-00625-f002:**
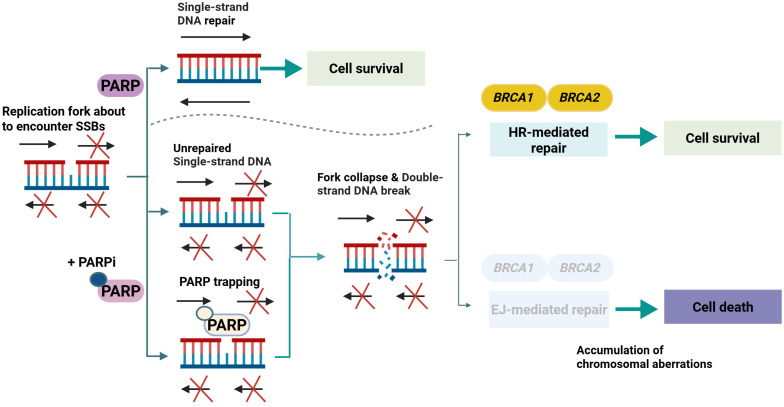
PARP Inhibitors and the Synthetic Lethality Paradigm. Initially, it was believed that PARP inhibitors worked by blocking PARylation and causing cytotoxicity. However, it was later discovered that the primary reason for tumor cell death was the trapping of the *PARP1* enzyme at DNA lesions. When DNA damage leads to SSBs, *PARP1* is responsible for their accurate repair. However, when *PARP1* becomes trapped, it poses a threat to replication forks during the S phase of the cell cycle. This ultimately results in the collapse of the replication fork and the creation of double-strand breaks. In cells with functional *BRCA* genes, homologous recombination (HR) can repair these breaks without errors. On the other hand, cells lacking *BRCA1/2* are deficient in HR and rely on error-prone DNA EJ pathways, such as classical non-homologous EJ or alternative EJ, to fix the double-strand breaks caused by the collapse of replication forks. This triggers the accumulation of chromosomal abnormalities and cell death through mitotic catastrophe. Abbreviations: *BRCA1/BRCA2*, breast cancer susceptibility gene 1/2; DSB, double-strand break; EJ, end-joining; HR, homologous recombination; PARP, poly (ADP-ribose) polymerase; PARPi, PARP inhibitor (poly (ADP-ribose) polymerase inhibitor); SSB, single-strand break. Created with Biorender (Accessed on 1 April 2025).

**Figure 3 biomolecules-15-00625-f003:**
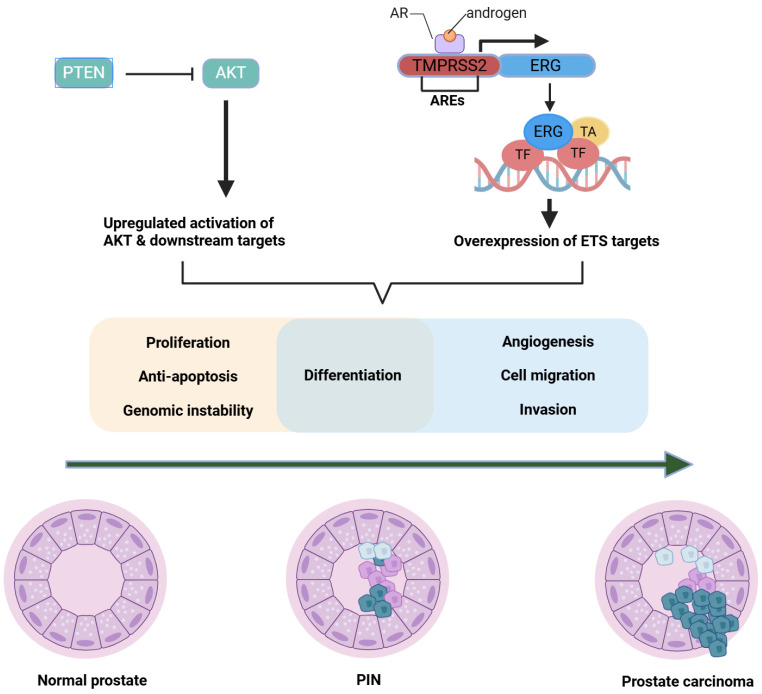
*TMPRSS2*–*ERG* Gene Fusion and AR-Driven Oncogenic Transcription. Loss of *PTEN* and concomitant activation of AKT could act in partnership with the *TMPRSS2*–*ERG* fusion protein to promote progression to prostate cancer through downstream pathways that increase the selective advantage of premalignant prostatic intraepithelial neoplasia (PIN) cells. Abbreviations: AKT, protein kinase B; AREs, androgen response elements; *ERG*, *ETS*-related gene; *ETS*, E26 transformation-specific; PIN, prostatic intraepithelial neoplasia; *PTEN*, phosphatase and tensin homolog; TA, transactivation domain; TF, transcription factor; *TMPRSS2*, transmembrane protease, serine 2. Created with Biorender (Accessed on 1 April 2025).

**Figure 4 biomolecules-15-00625-f004:**
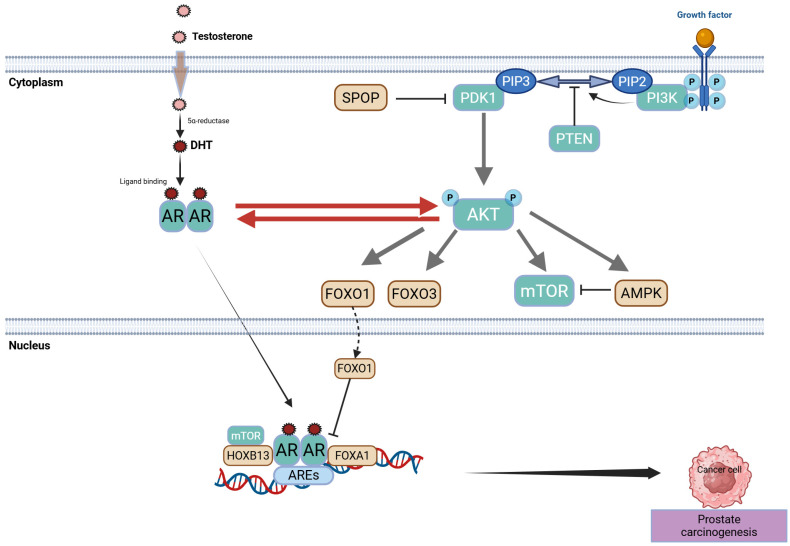
AR Axis and *PI3K/AKT/mTOR* Crosstalk. This schematic illustrates the reciprocal interactions between the AR signaling axis and the *PI3K/AKT/mTOR* pathway in prostate cancer. Testosterone is converted to DHT by 5α-reductase, and DHT-activated AR translocates into the nucleus to regulate gene transcription in collaboration with cofactors (e.g., *FOXA1, HOXB13*). Concurrently, growth factor signals drive PI3K-dependent production of PIP3 from PIP2, activating PDK1 and AKT. AKT phosphorylates multiple targets, including mTOR, promoting cell growth and survival. Tumor suppressors, such as *PTEN*, negatively regulate this pathway by converting PIP3 back to PIP2, while *SPOP* influences protein turnover within the pathway. Transcription factors *FOXO1* and *FOXO3* are regulated by both AR and AKT, integrating signals from each pathway. Crosstalk between AR and *PI3K/AKT/mTOR* signaling underpins key processes in prostate carcinogenesis, including proliferation, survival, and therapeutic resistance. Abbreviation: AKT, AKT serine/threonine kinase; AMPK, AMP-activated protein kinase; AR, androgen receptor; AREs, androgen response elements; DHT, dihydrotestosterone; *FOXA1*, forkhead box A1; *FOXO1*, forkhead box O1; *FOXO3*, forkhead box O3; *HOXB13*, homeobox B13; mTOR, mechanistic target of rapamycin (mTOR); PDK1, phosphoinositide-dependent kinase-1; PIP2, phosphatidylinositol 4,5-bisphosphate; PIP3, phosphatidylinositol (3,4,5)-trisphosphate; *PTEN*, phosphatase and tensin homolog; *SPOP*, speckle-type POZ protein. Created with Biorender (Accessed on 1 April 2025).

**Figure 5 biomolecules-15-00625-f005:**
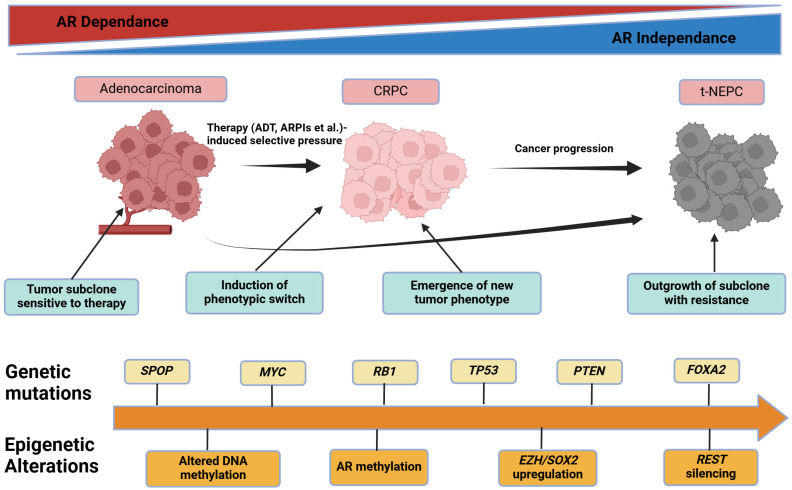
Tumor Heterogeneity, Clonal Evolution, and Treatment-Induced Neuroendocrine Prostate Cancer. This schematic illustrates how prostate adenocarcinoma evolves under the selective pressure of ADT and ARPIs, leading to the emergence of CRPC and, ultimately, t-NEPC. Initially, therapy-sensitive subclones may be outcompeted by therapy-resistant populations, driving a phenotypic switch and the appearance of new tumor subtypes. Over time, resistant subclones predominate, resulting in more aggressive disease states, such as NEPC. Underlying these transitions are key genetic mutations (e.g., in *SPOP*, *MYC*, *RB1*, *TP53*, *PTEN*, and *FOXA2*) and epigenetic alterations (e.g., altered DNA methylation, AR methylation, *EZH2/SOX2* upregulation, and *REST* silencing), which together promote tumor heterogeneity, clonal expansion, and therapeutic resistance. Abbreviations: ADT, androgen deprivation therapy; ARPI, androgen receptor pathway inhibitor; CRPC, castration-resistant prostate cancer; *MYC*, *MYC* proto-oncogene; *RB1*, retinoblastoma 1; *SPOP*, speckle-type POZ protein; t-NEPC, treatment induced neuroendocrine prostate cancer; *TP53*, tumor protein 53; *PTEN*, phosphatase and tensin homolog; *FOXA2*, forkhead box A2; *EZH2*, enhancer of zeste homolog 2 (histone methyltransferase); SOX2, sex determining region Y (SRY)-box transcription factor 2; *REST*, RE1 silencing transcription factor. Created with Biorender (Accessed on 1 April 2025).

**Figure 7 biomolecules-15-00625-f007:**
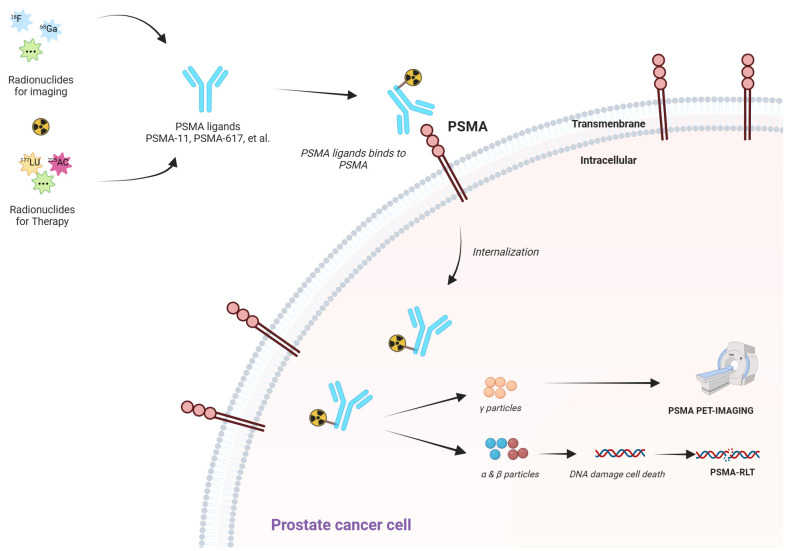
PSMA-Targeted Radioligand Therapy. This schematic illustrates the principle of PSMA-based imaging and therapy in prostate cancer. Radioligands (e.g., PSMA-11 for imaging and PSMA-617 for therapy) bind to the extracellular domain of PSMA on tumor cells. After binding, the complex undergoes internalization, delivering radioactive payloads into the cell. For imaging (PSMA-PET), positron-emitting isotopes enable visualization of tumor lesions, whereas therapeutic radionuclides, as used in PSMA-targeted radioligand therapy (PSMA-RLT), emit alpha or beta particles that induce DNA damage and cancer cell death. This targeted approach exploits high PSMA expression on prostate cancer cells, offering both diagnostic and therapeutic benefits. Abbreviation: APC, antigen-presenting cell; Arg-1, arginase-1; CAF, cancer-associated fibroblast; CD8^+^ T, CD8^+^ T cell; ECM, extracellular matrix; IFN-γ, interferon-gamma; IL-1, interleukin-1; IL-10, interleukin-10; M0-TAM, M0 tumor-associated macrophage; M1-TAM, M1 tumor-associated macrophage; M2-TAM, M2 tumor-associated macrophage; Mφ, macrophage; NF, normal fibroblast; NRG-1, neuregulin-1; PCa, prostate cancer; PD-L1, programmed death-ligand 1; PSMA, prostate-specific membrane antigen; RNS, reactive nitrogen species; ROS, reactive oxygen species; *SPP1*, secreted phosphoprotein 1; TGF-β, transforming growth factor-beta; TNF-α, tumor necrosis factor-alpha; Treg, regulatory T cell. Created with Biorender (Accessed on 1 April 2025).

**Table 1 biomolecules-15-00625-t001:** Major Molecular Alterations in Metastatic Prostate Cancer and Their Clinical Implications.

Molecular Target or Genetic Alteration	Key Mechanism/Function	Clinical Features	Clinical Utility
AR Amplification/AR Splice Variants (e.g., AR-*V7*)	Sustained AR signaling under low-androgen conditions; ligand-independent activation	Poor response or resistance to AR-targeted therapies; commonly seen in mCRPC	Predicts resistance to enzalutamide or abiraterone; potential biomarker for treatment selection
*PTEN* Loss	Hyperactivation of the *PI3K/AKT/mTOR* pathway; cross-talk with AR signaling	Associated with high-grade tumors and aggressive clinical course	May guide *PI3K/AKT/mTOR* inhibitor-based combination trials; potential prognostic indicator
DDR Defects (e.g., *BRCA2*, *ATM*)	Impaired DNA repair and increased genomic instability; vulnerability to *PARP* inhibition	More aggressive behavior if untreated; better response to PARP inhibitors	Companion diagnostics for PARP inhibitors; synthetic lethality-based therapy targeting
*TMPRSS2*–*ERG* Fusion	*ETS* transcription factor (*ERG*) overexpression; promotes invasion, EMT, and genomic instability	High prevalence in localized prostate cancer; variable association with outcomes in mPCa	Potential prognostic marker in combination with other alterations (e.g., *PTEN*)
*PI3K/AKT/mTOR Mutations*	Aberrant cell proliferation and survival; metabolic reprogramming	Often co-occurs with AR pathway alterations; contributes to therapeutic resistance	Under investigation in clinical trials targeting AKT and mTOR; potential combination strategy with AR inhibitors
*TP53/RB1* Co-mutations	Disruption of cell-cycle checkpoints; may facilitate lineage plasticity or neuroendocrine differentiation	Common in advanced mPCa; associated with poor prognosis	Emerging biomarker for early switch to chemotherapy or combination therapies

Abbreviations: AKT, protein kinase B; AR, androgen receptor; AR-*V7*, androgen receptor splice variant 7; *ATM*, ataxia telangiectasia mutated; *BRCA2*, breast cancer susceptibility gene 2; DDR, DNA damage repair; EMT, epithelial–mesenchymal transition; *ERG*, *ETS*-related gene; *ETS*, E26 transformation-specific; mCRPC, metastatic castration-resistant prostate cancer; mPCa, metastatic prostate cancer; mTOR, mechanistic target of rapamycin; PARP, poly (ADP-ribose) polymerase; PI3K, phosphoinositide 3-kinase; *PTEN*, phosphatase and tensin homolog; *RB1*, retinoblastoma 1; *TMPRSS2*, transmembrane protease, serine 2; *TP53*, tumor protein p53.

**Table 3 biomolecules-15-00625-t003:** Key Molecular Diagnostic Panels and Recommended Biomarkers in Prostate Cancer.

Diagnostic Panel/Biomarker	Testing Method	Clinical Significance	Limitations/Considerations
DDR-Focused Panel (*BRCA1/2, ATM*, etc.)	Targeted NGS or expanded gene panels Germline vs. somatic testing	Identifies candidates for PARP inhibitors and platinum-based therapies May inform familial genetic risk	Cost and limited access in some regions May miss epigenetic alterations
AR Splice Variants (e.g., AR-*V7*)	RT-PCR or ddPCR on CTCs Tissue-based RNA assays	Predicts resistance to enzalutamide or abiraterone Can guide switch to chemotherapy or other targeted agents	Variable sensitivity depending on assay Not yet universally available or standardized
*PTEN*/PI3K/AKT	IHC, FISH Targeted sequencing for hotspot mutation	Potential biomarker for AKT/mTOR inhibitors May correlate with disease aggressiveness	Limited predictive validation in some trials Reimbursement issues in certain regions
*TP53*/*RB1*	Targeted NGS or WES/WGS IHC for protein loss	Associated with poor prognosis May indicate early progression toward neuroendocrine differentiation	Rarely used in routine practice Data interpretation can be complex (co-occurring events)
*TMPRSS2*–*ERG* Fusion	FISH, RT-PCR, or RNA-seq	Possible prognostic marker when combined with other aberrations (e.g., *PTEN*)	Prognostic impact remains debated May not be actionable with current therapies

Abbreviations: AKT, protein kinase B; AR-*V7*, androgen receptor splice variant 7; *ATM*, ataxia telangiectasia mutated; *BRCA1*, breast cancer susceptibility gene 1; *BRCA2*, breast cancer susceptibility gene 2; CTCs, circulating tumor cells; DDR, DNA damage repair; ddPCR, droplet digital polymerase chain reaction; FISH, fluorescence in situ hybridization; IHC, immunohistochemistry; NGS, next-generation sequencing; PI3K, phosphoinositide 3-kinase; *PTEN*, phosphatase and tensin homolog; RT-PCR, reverse transcription polymerase chain reaction; *TMPRSS2*, transmembrane protease, serine 2; WES, whole exome sequencing; WGS, whole genome sequencing.

**Table 4 biomolecules-15-00625-t004:** Representative Performance Characteristics for Selected Molecular Biomarkers in Prostate Cancer.

Biomarker	Common Test Method (s)	Sensitivity (%)	Specificity (%)	PPV (%)	NPV (%)	Notes/References
AR-*V7* (Circulating Tumor Cells)	RT-PCR or ddPCR of isolated CTCs	~60–80	~70–90	~65–85	~70–90	Varies by platform and cutoff used. Higher PPV for resistance to AR inhibitors [104,108]. Not standardized across all labs.
*BRCA1/2* & DDR Panel (Genomic Testing)	NGS panels (targeted, WES)	~85–95	~95–99	~90–95	~85–95	Typically, high analytical accuracy for well-validated panels [6,89]. Clinical performance depends on presence of true driver mutations.
*PTEN* Loss (tissue-based)	IHC or FISH	~70–80	~80–90	~60–85	~75–90	Studies show variability in *PTEN* detection [52]. Co-occurring alterations (e.g., *TP53*) may influence true clinical impact.
*TP53*/*RB1* (tissue-based)	Targeted NGS, WES/WGS, IHC for protein	~70–90	~85–95	~80	~85	Highly dependent on sample purity and co-mutations [109]; associated with poor prognosis
Mismatch Repair Defects (e.g., *MSH2, MLH1*)	IHC for mismatch repair proteins, NGS	~85–95	~85–95	~75–90	~90	Predictive for PD-1 checkpoint inhibition; modest incidence in PCa [6]
*TMPRSS2*–*ERG* Fusion (Tissue-Based Assays)	FISH, RT-PCR, or RNA-seq in tumor tissue	~50–70	~80–95	~70–90	~60–80	Variable positivity rates in localized vs. metastatic PCa [49,56]. Prognostic value still debated.

Abbreviations: AKT, AKT serine/threonine kinase; AR, androgen receptor; AR-*V7*, androgen receptor splice variant 7; BiTE, bispecific T cell engager; *BRCA1/2*, breast cancer susceptibility gene 1/2; CTC, circulating tumor cell; DDR, DNA damage repair; dMMR, deficient mismatch repair; FISH, fluorescence in situ hybridization; GI, gastrointestinal; HR, homologous recombination; HRR, homologous recombination repair; IHC, immunohistochemistry; MSI-H, microsatellite instability-high; mCRPC, metastatic castration-resistant prostate cancer; mCSPC, metastatic castration-sensitive prostate cancer; NGS, next-generation sequencing; OS, overall survival; PARP, poly (ADP-ribose) polymerase; PCa, prostate cancer; PD-1, programmed death-1; PD-L1, programmed death ligand-1; PET, positron emission tomography; PSMA, prostate-specific membrane antigen; *PTEN*, phosphatase and tensin homolog; rPFS, radiographic progression-free survival; RT-PCR, reverse transcription polymerase chain reaction; ddPCR, droplet digital polymerase chain reaction; *STEAP1*, six-transmembrane epithelial antigen of prostate 1; TMB, tumor mutational burden; TME, tumor microenvironment; *TP53*, tumor protein p53; *RB1*, retinoblastoma 1; WES, whole exome sequencing; WGS, whole genome sequencing; RNA-seq, RNA sequencing; *MSH2*, MutS homolog 2; *MLH1*, MutL homolog 1; *TMPRSS2*, transmembrane protease serine 2; *ERG*, *ETS*-related gene.

**Table 5 biomolecules-15-00625-t005:** Liquid Biopsy Modalities in Metastatic Prostate Cancer: Key Features and Clinical Applications.

Modality	Specimen Characteristics	Analytical Techniques	Clinical Applications	Advantages	Limitations
ctDNA	Cell-free DNA fragments shed by tumor cells Detected in plasma or serum	Targeted/whole-exome NGS ddPCR	Real-time monitoring of tumor burden Detection of actionable mutations (e.g., *BRCA2*)	Minimally invasive Repeat sampling feasible Reflects genomic heterogeneity	Low abundance in early disease Sensitivity depends on tumor fraction Assay costs and standardization issues
CTCs	Intact, viable tumor cells in the bloodstream May be enriched via immunomagnetic or size-based separation methods	Immunophenotyping Single-cell genomics/transcriptomics	Prognostic biomarker (CTC count) AR-*V7* status for therapy guidance Potential ex vivo drug testing	Allows morphological and molecular analyses Provides insight into specific cell populations	Rare cells, labor-intensive Limited sensitivity in low-volume disease Heterogeneity among different CTC populations
Exosomes and Extracellular Vesicles	Nano-scale vesicles containing proteins, RNA, and DNA Released by tumor and stromal cells into bodily fluids	RNA-seq, proteomics Nanoparticle tracking Advanced mass spectrometry	May reveal early resistance signatures Potential biomarkers for immune and stromal interactions	Reflects active secretory pathways Can capture tumor–stromal communication	Isolation protocols not standardized Complexity of vesicle subtypes Data interpretation is challenging

Abbreviations: AR-*V7*, androgen receptor splice variant 7; *BRCA2*, breast cancer 2; CTCs, circulating tumor cells; ddPCR, digital droplet polymerase chain reaction; exosomes, extracellular Vesicles; NGS, next-generation sequencing; PCR, polymerase chain reaction; RNA-seq, RNA sequencing; ddPCR, digital droplet PCR.

**Table 6 biomolecules-15-00625-t006:** Major Clinical Trials of Targeted and Immunotherapeutic Approaches in Prostate Cancer.

Treatment or Combination	Primary Target/Mechanism	Trial Phase	Patient Population	Key Outcomes	Current Status	Reference
Olaparib vs. Abiraterone/Enzalutamide (PROfound)	*PARP* inhibition (DDR deficiency)	Phase III	mCRPC with HRR gene alterations (e.g., *BRCA1/2*)	Improved radiographic PFS and OS in biomarker-selected patients	Approved for HRR-mutated mCRPC	[175]
Ipatasertib + Abiraterone (IPATential150)	AKT inhibitor + AR axis blockade	Phase III	mCRPC, particularly with *PTEN* loss	Prolonged PFS in the *PTEN*-loss subgroup	Ongoing or completed; subset analyses continuing	[151]
177Lu-PSMA-617 + Standard of Care (VISION)	PSMA-targeted radioligand therapy	Phase III	Heavily pretreated mCRPC	Improved OS and PFS vs. standard care	Approved in multiple regions	[176]
Nivolumab + Ipilimumab (CheckMate 650)	Dual immune checkpoint blockade (PD-1, CTLA-4)	Phase II	mCRPC, previously treated	Moderate objective response; significant immune-related toxicity	Further refinement of combination strategies needed	[177]
Pembrolizumab (KEYNOTE-199)	PD-1 immune checkpoint blockade	Phase II	mCRPC with prior treatments	Modest response rates; better outcomes in MSI-H or DNA repair defects	Investigational in selected biomarker-defined subgroups	[178]
Apalutamide (SPARTAN)	Next-generation AR antagonist	Phase III	nmCRPC (non-metastatic CRPC)	Significantly improved metastasis-free survival (MFS)	Approved for nmCRPC	[179]

Abbreviations: AKT, AKT serine/threonine kinase; AR, androgen receptor; *BRCA1/2*, breast cancer susceptibility genes 1 and 2; CTLA-4, cytotoxic T-lymphocyte antigen 4; DDR, DNA damage repair; HRR, homologous recombination repair; mCRPC, metastatic castration-resistant prostate cancer; MFS, metastasis-free survival; MSI-H, microsatellite instability-high; OS, overall survival; PARP, poly (ADP-ribose) polymerase; PD-1, programmed death-1; PFS, progression-free survival; PSMA, prostate-specific membrane antigen; *PTEN*, phosphatase and tensin homolog.

**Table 7 biomolecules-15-00625-t007:** Key Therapeutic Strategies in Advanced Prostate Cancer: Mechanisms, Biomarkers, and Clinical Considerations.

Therapeutic Strategy	Mechanism of Action	Associated Biomarkers	Advantages	Disadvantages/Limitations	Reference
AR-Targeted Therapies (e.g., Abiraterone, Enzalutamide, Apalutamide, Darolutamide)	Second-generation agents block AR function by inhibiting ligand binding (enzalutamide) or androgen biosynthesis (abiraterone)	AR-*V7* and other splice variants AR amplification, LBD mutations	Mainstay for mCSPC/mCRPC with proven OS benefits Generally well-tolerated compared to chemotherapy Can be combined with other targeted agents	Inevitable resistance (AR mutations/splice variants) Cross-resistance among agents Limited efficacy in NEPC	[29,136,137,140,141,142,143]
PARP Inhibitors (e.g., Olaparib, Rucaparib)	Exploit synthetic lethality in tumors with deficient DNA repair (e.g., *BRCA2*)	*BRCA1/2, ATM, PALB2*, or other DDR gene alterations	Markedly improved outcomes in *BRCA2*-mutant mCRPC Potential synergy with AR pathway inhibition or chemo Established companion diagnostics for biomarker-driven selection	Resistance via reversion mutations or upregulation of alternative repair pathways Limited efficacy in non-*BRCA/ATM*-altered tumors Class-related toxicities (myelosuppression, GI events) High cost and variable global availability	[46,146,147]
Immunotherapy (Checkpoint Inhibitors, Vaccines, Bispecific T cell Engagers)	Enhances immune-mediated tumor recognition and destruction (e.g., PD-1/PD-L1 or CTLA-4 blockade) BiTEs (e.g., *STEAP1*-targeting) redirect T cells to tumor cells Vaccines stimulate specific T cell responses	MMR deficiency, MSI-H Emerging markers (e.g., tumor mutation burden, PD-L1 expression, *STEAP1* expression)	Durable responses in a subset of patients with high TMB/MSI-H Potential synergy with radiation or targeted therapies Bispecific T cell engagers can be highly potent, bypassing the need for prior T cell priming	Prostate cancer is typically immunologically “cold” Limited responses outside biomarker-selected populations Immune-related adverse events (e.g., colitis, dermatitis) BiTE therapies can cause cytokine release syndrome	[154,155,156,157,158,159]
PSMA-Targeted Radioligand Therapy (e.g., ^177^Lu-PSMA-617)	Delivers cytotoxic radiation specifically to PSMA-expressing tumor cells	PSMA PET imaging for target expression	Offers both diagnostic (PSMA PET-CT) and therapeutic potential (“theranostics”) Demonstrated survival advantage in heavily pretreated mCRPC (VISION trial) Can target micro-metastatic disease	Not all PCa lesions overexpress PSMA (e.g., neuroendocrine variants) Salivary gland toxicity and xerostomia, especially with alpha-emitters (225Ac) Requires specialized facilities for radiopharmaceutical handling Cost and limited access in some regions	[161,162,167,169,171]
*PI3K/AKT/mTOR* Inhibitors (e.g., Ipatasertib, Capivasertib)	Blocks downstream signaling of *PI3K/AKT/mTOR* axis, often hyperactivated due to *PTEN* loss	*PTEN* status *PIK3CA,* AKT mutations	Potential synergy with AR blockade May delay disease progression in *PTEN*-deficient mCRPC Encouraging early-phase trial results in combination regimens	On-target metabolic toxicities (hyperglycemia, rash) Relatively modest single-agent activity Patient selection crucial to avoid unnecessary toxicity	[150,151,152,153]

Abbreviations: AKT, AKT serine/threonine kinase; AR, androgen receptor; AR-*V7*, androgen receptor splice variant 7; BiTE, bispecific T cell engager; *BRCA1/2*, breast cancer susceptibility gene 1/2; CTC, circulating tumor cell; DDR, DNA damage repair; dMMR, deficient mismatch repair; GI, gastrointestinal; HR, homologous recombination; HRR, homologous recombination repair; LBD, ligand-binding domain; MSI-H, microsatellite instability-high; mCRPC, metastatic castration-resistant prostate cancer; mCSPC, metastatic castration-sensitive prostate cancer; NGS, next-generation sequencing; OS, overall survival; PARP, poly (ADP-ribose) polymerase; PCa, prostate cancer; PD-1, programmed death-1; PD-L1, programmed death ligand-1; PET, positron emission tomography; PSMA, prostate-specific membrane antigen; *PTEN*, phosphatase and tensin homolog; rPFS, radiographic progression-free survival; *STEAP1*, six-transmembrane epithelial antigen of prostate 1; TMB, tumor mutational burden; TME, tumor microenvironment.

## Data Availability

No new data were created or analyzed in this study. Data sharing is not applicable to this article.

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
