# Peer review of "Precision Targeting in Metastatic Prostate Cancer: Molecular Insights to Therapeutic Frontiers"

_biomolecules, 2025, doi:10.3390/biom15050625_

Round 1
Reviewer 1 Report
Comments and Suggestions for Authors
The manuscript is quite interesting, however the authors need to address several issues to further ameliorate it:
- it could be useful to include a comparative table summarizing the key therapeutic strategies (e.g., PARP inhibitors, PSMA therapy, immunotherapy) with their respective advantages, disadvantages, and associated biomarkers.
- Some references are outdated, particularly those concerning emerging biomarkers and the use of liquid biopsy for monitoring AR-V7. Update citations by including more recent studies
- Add a table containing data on the sensitivity, specificity, and positive/negative predictive value of molecular biomarkers
Comments on the Quality of English Language
- Some sentences are lengthy and complex and there are some grammatical and punctuation errors. Consider submitting the manuscript for language editing by a native English speaker for enhanced clarity.
Author Response
"Please see the attachment."

Reviewer 2 Report
Comments and Suggestions for Authors
This manuscript provides a comprehensive review of the mechanisms leading to prostate cancer, the physiology underlying the development of resistance to therapies, in particular androgen therapy, ongoing therapeutic approaches, and future directions. I have only a few suggestions that the authors might want to consider for inclusion.
- The authors discuss the value of PET/CT PSMA imaging, and point out the limitation that many centers may not have the capability to perform PET/CT imaging. To circumvent this shortcoming, a number of [99Tm] tracers have been developed for use with the more widely available SPECT/CT technology. A recent meta-analysis (Wang, Ketteler et al, BMC Cancer 2024) supports the potential of this approach.
- In addition to the potential of replacing lutetium with alpha emitters in PSMA-directed theranostics, the beta-emitter terbium, [161Tb]Tb-PSMA-I&T is being tested in the Phase I/II VIOLET (Bureau et al, J Nucl Med 2024). [161Tb] emits Auger and conversion electrons that deposit a higher concentration of radiation over a shorter path, and shows superior in vitro and in vivo efficacy compared to 177Lu.
- Bipolar androgen therapy (BAT), the monthly administration of high dose testosterone to ADT-treated patients, has shown efficacy in patients progressing on abiraterone (TRANSFORMER, Denmeade et al, J Clin Oncol 2021). BAT provided meaningful clinical efficacy, and also sensitized patients to subsequent Enzalutamide treatment. A more recent Phase 2 study (COMBAT) has also shown efficacy of BAT plus Nivolumab, and found that BAT induced pro-inflammatory gene expression changes in metastatic tumor biopsies that were restricted to patients achieving a clinical response (Markowski et al Nat Comm 2024).
Author Response
"Please see the attachment."
